# On the (In)fidelity and Sensitivity of Explanations

**Chih-Kuan Yeh** [*], **Cheng-Yu Hsieh** [†], **Arun Sai Suggala** [‡]
Department of Machine Learning
Carnegie Mellon University

**David I. Inouye** [§]
School of Electrical and Computer Engineering
Purdue University

**Pradeep Ravikumar** [¶]
Department of Machine Learning
Carnegie Mellon University

## Abstract

We consider objective evaluation measures of saliency explanations for complex black-box machine learning models. We propose simple robust variants of two notions that have been considered in recent literature: (in)fidelity, and sensitivity. We analyze optimal explanations with respect to both these measures, and while the optimal explanation for sensitivity is a vacuous constant explanation, the optimal explanation for infidelity is a novel combination of two popular explanation methods. By varying the perturbation distribution that defines infidelity, we obtain novel explanations by optimizing infidelity, which we show to out-perform existing explanations in both quantitative and qualitative measurements. Another salient question given these measures is how to modify *any given explanation* to have better values with respect to these measures. We propose a simple modification based on lowering sensitivity, and moreover show that when done appropriately, we could simultaneously improve both sensitivity as well as fidelity.

## 1 Introduction

We consider the task of how to explain a complex machine learning model, abstracted as a function that predicts a response given an input feature vector, given only black-box access to the model. A popular approach to do so is to attribute any given prediction to the set of input features: ranging from providing a vector of importance weights, one per input feature, to simply providing a set of important features. For instance, given a deep neural network for image classification, we may explain a specific prediction by showing the set of salient pixels, or a heatmap image showing the importance weights for all the pixels. But how good is any such explanation mechanism? We can distinguish between two classes of explanation evaluation measures [22, 27]: objective measures and subjective measures. The predominant evaluations of explanations have been subjective measures, since the notion of explanation is very human-centric; these range from qualitative displays of explanation examples, to crowd-sourced evaluations of human satisfaction with the explanations, as well as whether humans are able to understand the model. Nonetheless, it is also important to consider objective measures of explanation effectiveness, not only because these place explanations on a sounder theoretical foundation, but also because they allow us to *improve* our explanations by improving their objective measures.

One way to objectively evaluate explanations is to verify whether the explanation mechanism satisfies (or does not satisfy) certain axioms, or properties [25, 43]. In this paper, we focus on quantitative objective measures, and provide and analyze two such measures. First, we formalize the notion of fidelity of an explanation to the predictor function. One natural approach to measure fidelity, when we have apriori information that only a particular subset of features is relevant, is to test if the

---

[*] cjyeh@cs.cmu.edu

[†] chyu.hsieh@gmail.com

[‡] asuggala@andrew.cmu.edu

[§] dinouye@purdue.edu

[¶] pradeepr@cs.cmu.edu

features with high explanation weights belong to this relevant subset [10]. In the absence of such apriori information, Ancona et al. [3] provide a more quantitative perspective on the earlier notion by measuring the correlation between the sum of a subset of feature importances and the difference in function value when setting the features in the subset to some reference value; by varying the subsets, we get different values of such subset correlations. In this work, we consider a simple generalization of this notion, that produces a single fidelity measure, which we call the infidelity measure.

Our infidelity measure is defined as the expected difference between the two terms: (a) the dot product of the input perturbation to the explanation and (b) the output perturbation (i.e., the difference in function values after *significant perturbations* on the input). This general setup allows for a varied class of significant perturbations: a non-random perturbation towards a single reference or baseline value, perturbations towards multiple reference points e.g. by varying subsets of features to perturb, and a random perturbation towards a reference point with added small Gaussian noise, which allows the infidelity measure to be robust to small mis-specifications or noise in either the test input or the reference point.

We then show that the optimal explanation that minimizes this infidelity measure could be loosely cast as a novel combination of two well-known explanation mechanisms: Smooth-Grad [40] and Integrated Gradients [43] using a kernel function specified by the random perturbations. As another validation of our formalization, we show that many recently proposed explanations can be seen as optimal explanations for the infidelity measure with specific perturbations. We also introduce new perturbations which lead to novel explanations by optimizing the infidelity measure, and we validate the explanations are qualitatively better through human experiments. It is worth emphasizing that the infidelity measure, while objective, may not capture all the desiderata of a successful explanation; thus, it is still of interest to take a given explanation that does not have the form of the optimal explanation with respect to a specified infidelity measure and *modify it* to have lesser infidelity.

Analyzing this question leads us to another objective measure: the sensitivity of an explanation, which measures the degree to which the explanation is affected by insignificant perturbations from the test point. It is natural to wish for our explanation to have low sensitivity, since that would entail differing explanations with minor variations in the input (or prediction values), which might lead us to distrust the explanations. Explanations with high sensitivity could also be more amenable to adversarial attacks, as Ghorbani et al. [13] show in the context of gradient based explanations. Regardless, we largely expect explanations to be simple, and lower sensitivity could be viewed as one such notion of simplicity. Due in part to this, there have been some recent efforts to quantify the sensitivity of explanations [2, 28, 13]. We propose and analyze a simple robust variant of these recent proposals that is amenable to Monte Carlo sampling-based approximation. Our key contribution, however, is in relating the notion of sensitivity to our proposed notion of infidelity, which also allows us to address the earlier raised question of how to modify an explanation to have better fidelity. Asking this question for sensitivity might seem vacuous, since the optimal explanation that minimizes sensitivity (for all its related variants) is simply a trivial constant explanation, which is naturally not a desired explanation. So a more interesting question would be: how do we modify a given explanation so that it has lower sensitivity, but not too much. To quantify the latter, we could in turn use fidelity.

As one key contribution of the paper, we show that a restrained lowering of the sensitivity of an explanation also *increases* its fidelity. In particular, we consider a simple kernel smoothing based algorithm that appropriately lowers the sensitivity of any given explanation, but importantly also lowers its infidelity. Our meta-algorithm encompasses Smooth-Grad [40] which too modifies any existing explanation mechanism by averaging explanations in a small local neighborhood of the test point. In the appendix, we also consider an alternative approach to improve gradient explanation sensitivity and fidelity by adversarial training, which however requires that we be able to modify the given predictor function itself, which might not always be feasible. Our modifications improve both sensitivity and fidelity in most cases, and also provides explanations that are qualitatively better, which we validate in a series of experiments.[6]

## 2   Objective Measure: Explanation Infidelity

Consider the following general supervised learning setting: input space $\mathcal{X} \subseteq \mathbb{R}^d$, an output space $\mathcal{Y} \subseteq \mathbb{R}$, and a (machine-learnt) black-box predictor $\mathbf{f} : \mathbb{R}^d \mapsto \mathbb{R}$, which at some test input $\mathbf{x} \in \mathbb{R}^d$, predicts the output $\mathbf{f}(\mathbf{x})$. Then a feature attribution explanation is some function $\Phi : \mathcal{F} \times \mathbb{R}^d \mapsto \mathbb{R}^d$, that given a black-box predictor $\mathbf{f}$, and a test point $\mathbf{x}$, provides importance scores $\Phi(\mathbf{f}, \mathbf{x})$ for the set of

input features. We let $\| \cdot \|$ denote a given norm over the input and explanation space. In experiments, if not specified, this will be set to the $\ell_2$ norm.

## 2.1 Defining the infidelity measure

A natural notion of the goodness of an explanation is to quantify the degree to which it captures how the predictor function itself changes in response to significant perturbations. Along this spirit, [4, 37, 43] propose the completeness axiom for explanations consisting of feature importances, which states that the sum of the feature importances should sum up to the difference in the predictor function value at the given input and some specific baseline. [3] extend this to require that the sum of a subset of feature importance weights should sum up to the difference in the predictor function value at the given input and to a perturbed input that sets the subset of features to some specific baseline value. When the subset of features is large, this would entail that explanations capture the combined importance of the subset of features even if not the individual feature importances, and when the the subset of features is small, this would entail that explanations capture the individual importance of features. We note that this can be contrasted with requiring the explanations to capture the function values itself as in the causal local explanation metric of [30], rather than the difference in function values, but we focus on the latter. Letting $S_k$ denote a subset of $k$ features, [3] measured the discrepancy of the above desiderata as the correlation between $\sum_{i \in S_k} \Phi(\mathbf{f}, \mathbf{x})_i$ and $\mathbf{f}(\mathbf{x}) - \mathbf{f}(\mathbf{x}[\mathbf{x}_{S_k} = 0])$, where $\mathbf{x}[\mathbf{x}_S = a]_j = a\,\mathbb{I}(j \in S) + \mathbf{x}_j\,\mathbb{I}(j \notin S)$ and $\mathbb{I}$ is the indicator function.

One minor caveat with the above is that we may be interested in perturbations more general than setting feature values to 0, or even to a single baseline; for instance, we might simultaneously require smaller discrepancy over a set of subsets, or some distribution of subsets (as is common in game theoretic approaches to deriving feature importances [11, 42, 25]), or even simply a prior distribution over the baseline input. The correlation measure also focuses on second order moments, and is not as easy to optimize. We thus build on the above developments, by first allowing random perturbations on feature values instead of setting certain features to some baseline value, and secondly, by replacing correlation with expected mean square error (our development could be further generalized to allow for more general loss functions). We term our evaluation measure *explanation infidelity*.

**Definition 2.1.** Given a black-box function $\mathbf{f}$, explanation functional $\Phi$, a random variable $\mathbf{I} \in \mathbb{R}^d$ with probability measure $\mu_{\mathbf{I}}$, which represents meaningful perturbations of interest, we define the explanation infidelity of $\Phi$ as:

$$\text{INFD}(\Phi, \mathbf{f}, \mathbf{x}) = \mathbb{E}_{\mathbf{I} \sim \mu_{\mathbf{I}}} \left[ \left( \mathbf{I}^T \Phi(\mathbf{f}, \mathbf{x}) - (\mathbf{f}(\mathbf{x}) - \mathbf{f}(\mathbf{x} - \mathbf{I})) \right)^2 \right]. \tag{1}$$

$\mathbf{I}$ represents significant perturbations around $\mathbf{x}$, and can be specified in various ways. We begin by listing various plausible perturbations of interest.

- Difference to baseline: $\mathbf{I} = \mathbf{x} - \mathbf{x}_0$, the difference between input and baseline.
- Subset of difference to baseline: for any fixed subset $S_k \subseteq [d]$, $\mathbf{I}_{S_k} = \mathbf{x} - \mathbf{x}[\mathbf{x}_{S_k} = (\mathbf{x}_0)_{S_k}]$ corresponds to the perturbation in the correlation measure of [3] when $\mathbf{x}_0 = 0$.
- Difference to noisy baseline: $\mathbf{I} = \mathbf{x} - \mathbf{z}_0$, where $\mathbf{z}_0 = \mathbf{x}_0 + \epsilon$, for some zero mean random vector $\epsilon$, for instance $\epsilon \sim \mathcal{N}(0, \sigma^2)$.
- Difference to multiple baselines: $\mathbf{I} = \mathbf{x} - \mathbf{x}_0$, where $\mathbf{x}_0$ is a random variable that can take multiple values.

As we will next show in Section 2.3, many recently proposed explanations could be viewed as optimizing the aforementioned infidelity for varying perturbations $\mathbf{I}$. Our proposed infidelity measurement can thus be seen as a unifying framework for these explanations, but moreover, as a way to design new explanations, and evaluate any existing explanations.

## 2.2 Explanations with least Infidelity

Given our notion of infidelity, a natural question is: what is the explanation that is optimal with respect to infidelity, that is, has the least infidelity possible. This naturally depends on the distribution of the perturbations $\mathbf{I}$, and its surprisingly simple form is detailed in the following proposition.

**Proposition 2.1.** *Suppose the perturbations* $\mathbf{I}$ *are such that* $\left( \int \mathbf{I}\mathbf{I}^T d\mu_{\mathbf{I}} \right)^{-1}$ *is invertible. The optimal explanation* $\Phi^*(\mathbf{f}, \mathbf{x})$ *that minimizes infidelity for perturbations* $\mathbf{I}$ *can then be written as*

$$\Phi^*(\mathbf{f}, \mathbf{x}) = \left( \int \mathbf{I}\mathbf{I}^T d\mu_{\mathbf{I}} \right)^{-1} \left( \int \mathbf{I}\mathbf{I}^T IG(\mathbf{f}, \mathbf{x}, \mathbf{I}) d\mu_{\mathbf{I}} \right), \tag{2}$$

where $\text{IG}(\mathbf{f}, \mathbf{x}, \mathbf{I}) = \int_{t=0}^{1} \nabla \mathbf{f}(\mathbf{x} + (t-1)\mathbf{I}) \, dt$ is the integrated gradient of $\mathbf{f}(\cdot)$ between $(\mathbf{x} - \mathbf{I})$ and $\mathbf{x}$ [43], but can be replaced by any functional that satisfies $\mathbf{I}^T \text{IG}(\mathbf{f}, \mathbf{x}, \mathbf{I}) = \mathbf{f}(\mathbf{x}) - \mathbf{f}(\mathbf{x} - \mathbf{I})$. A generalized version of SmoothGrad can be written as $\Phi_k(\mathbf{f}, \mathbf{x}) := [\int_{\mathbf{z}} k(\mathbf{x}, \mathbf{z})]^{-1} \int_{\mathbf{z}} \Phi(\mathbf{f}, \mathbf{z}) \, k(\mathbf{x}, \mathbf{z}) d\mathbf{z}$ where the Gaussian kernel can be replaced by any kernel. Therefore, the optimal solution of Proposition 2.1 can be seen as applying a smoothing operation reminiscent of SmoothGrad on Integrated Gradients (or any explanation that satisfies the completeness axiom), where a special kernel $\mathbf{I}\mathbf{I}^T$ is used instead of the original kernel $k(\mathbf{x}, \mathbf{z})$. When I is deterministic, the integral of $\mathbf{I}\mathbf{I}^T$ is rank-one and cannot be inverted, but being optimal with respect to the infidelity can be shown to be equivalent to satisfying the Completeness Axiom. To enhance computational stability, we can replace inverse by pseudo-inverse, or add a small diagonal matrix to overcome the non-invertible case, which works well in experiments.

## 2.3 Many Recent Explanations Optimize Infidelity

As we show in the sequel, many recently proposed explanation methods can be shown to be optimal with respect to our infidelity measure in Definition 2.1, for varying perturbations $\mathbf{I}$.

**Proposition 2.2.** *Suppose the perturbation $\mathbf{I} = \mathbf{x} - \mathbf{x}_0$ is deterministic and is equal to the difference between $\mathbf{x}$ and some baseline $\mathbf{x}_0$. Let $\Phi^*(\mathbf{f}, \mathbf{x})$ be any explanation which is optimal with respect to infidelity for perturbations $\mathbf{I}$. Then $\Phi^*(\mathbf{f}, \mathbf{x}) \odot \mathbf{I}$ satisfies the completeness axiom; that is $\sum_{j=1}^{d} [\Phi^*(\mathbf{f}, \mathbf{x}) \odot \mathbf{I}]_j = \mathbf{f}(\mathbf{x}) - \mathbf{f}(\mathbf{x} - \mathbf{I})$. Note that the completeness axiom is also satisfied by IG [43], DeepLift [37], LRP [4].*

**Proposition 2.3.** *Suppose the perturbation is given by $\mathbf{I}_\epsilon = \epsilon \cdot \mathbf{e}_i$, where $\mathbf{e}_i$ is a coordinate basis vector. Then the optimal explanation $\Phi_\epsilon^*(\mathbf{f}, \mathbf{x})$ with respect to infidelity for perturbations $\mathbf{I}_\epsilon$, satisfies: $\lim_{\epsilon \to 0} \Phi_\epsilon^*(\mathbf{f}, \mathbf{x}) = \nabla \mathbf{f}(\mathbf{x})$, so that the limit point of the optimal explanations is the gradient explanation [36].*

**Proposition 2.4.** *Suppose the perturbation is given by $\mathbf{I} = \mathbf{e}_i \odot \mathbf{x}$, where $\mathbf{e}_i$ is a coordinate basis vector. Let $\Phi^*(\mathbf{f}, \mathbf{x})$ be the optimal explanation with respect to infidelity for perturbations $\mathbf{I}$. Then $\Phi^*(\mathbf{f}, \mathbf{x}) \odot \mathbf{x}$ is the occlusion-1 explanation[47].*

**Proposition 2.5.** *Following the notation in [25], given a test input $\mathbf{x}$, suppose there is a mapping $h_\mathbf{x} : \{0, 1\}^d \mapsto \mathbb{R}^d$ that maps simplified binary inputs $\mathbf{z} \in \{0, 1\}^d$ to $\mathbb{R}^d$, such that the given test input $\mathbf{x}$ is equal to $h_\mathbf{x}(\mathbf{z}_0)$ where $\mathbf{z}_0$ is a vector with all ones and $h_\mathbf{x}(\mathbf{0}) = 0$ where $\mathbf{0}$ is the zero vector. Now, consider the perturbation $\mathbf{I} = h_\mathbf{x}(\mathbf{Z})$, where $\mathbf{Z} \in \{0, 1\}^d$ is a binary random vector with distribution $\mathbb{P}(\mathbf{Z} = \mathbf{z}) \propto \frac{d-1}{\binom{d}{\|\mathbf{z}\|_1} \|\mathbf{z}\|_1 (d - \|\mathbf{z}\|_1)}$. Then for the optimal explanation $\Phi^*(\mathbf{f}, \mathbf{x})$ with respect to infidelity for perturbations $\mathbf{I}$, $\Phi^*(\mathbf{f}, \mathbf{x}) \odot \mathbf{x}$ is the Shapley value[25].*

## 2.4 Some Novel Explanations with New Perturbations

By varying the perturbations $\mathbf{I}$ in our infidelity definition 2.1, we not only recover existing explanations (as those that optimize the corresponding infidelity), but also design some novel explanations. We provide two such instances below.

**Noisy Baseline.** The completeness axiom is one of the most commonly adopted axioms in the context of explanations, but a caveat is that the baseline is set to some fixed vector, which does not account for noise in the input (or the baseline itself). We thus set the baseline to be a Gaussian random vector centered around a certain clean baseline (such as the mean input or zero) depending on the context. The explanation that optimizes infidelity with corresponding perturbations $\mathbf{I}$ is a novel explanation that can be seen as satisfying a robust variant of the completeness axiom.

**Square Removal.** Our second example is specific for image data. We argue that perturbations that remove random subsets of pixels in images may be somewhat meaningless, since there is very little loss of information given surrounding pixels that are not removed. Also ranging over all possible subsets to remove (as in SHAP [25]) is infeasible for high dimension images. We thus propose a modified subset distribution from that described in Proposition 2.5 where the perturbation $\mathbf{Z}$ has a uniform distribution over square patches with predefined length, which is in spirit similar to the work of [49]. This not only improves the computational complexity, but also better captures spatial relationships in the images. One can also replace the square with more complex random masks designed specifically for the image domain [29].

## 2.5 Local and Global Explanations

As discussed in [3], we can contrast between local and global feature attribution explanations: global feature attribution methods directly provide the change in the function value given changes in the features, whereas local feature attribution methods focus on the sensitivity of the function to the changes to the features, so that the local feature attributions need to be multiplied with the input to obtain an estimate of the change in the function value. Thus, for gradient-based explanations considered in [3], the raw explanation such as the gradient itself is a local explanation, while the raw explanation multiplied with the raw input is called a global explanation. In our context, explanations optimizing Definition 2.1 are naturally local explanations as $\mathbf{I}$ is real-valued. However, this can be easily modified to a global explanation by multiplying with $\mathbf{x} - \mathbf{x}_0$ when $\mathbf{I}$ is a subset of $\mathbf{x} - \mathbf{x}_0$. The reason we emphasize this distinction is that since global and local explanations capture subtly different aspects, they should be compared separately. We note that our definition of local and global explanations follows the description of [3], distinct from that in [30].

## 3 Objective Measure: Explanation Sensitivity

A classical approach to measure the sensitivity of a function, would simply be the gradient of the function with respect to the input. Therefore, the sensitivity of the explanation can be defined as: for any $j \in \{1, \ldots, d\}$,

$$[\nabla_{\mathbf{x}} \Phi(\mathbf{f}(\mathbf{x}))]_j = \lim_{\epsilon \to 0} \frac{\Phi(\mathbf{f}(\mathbf{x} + \epsilon\,\mathbf{e}_j)) - \Phi(\mathbf{f}(\mathbf{x}))}{\epsilon},$$

where $\mathbf{e}_j \in \mathbb{R}^d$ is the $j$-th coordinate basis vector, with $j$-th entry one and all others zero. It quantifies how the explanation changes as the input is varied infinitesimally. And as a scalar-valued summary of this sensitivity vector, a natural approach is to simply compute some norm of the sensitivity matrix: $\|\nabla_{\mathbf{x}} \Phi(\mathbf{f}(\mathbf{x}))\|$. A slightly more robust variant would be a locally uniform bound:

$$\text{SENS}_{\text{GRAD}}(\Phi, \mathbf{f}, \mathbf{x}, r) = \sup_{\|\delta\| \leqslant r} \|\nabla_{\mathbf{x}} \Phi(\mathbf{x} + \delta)\|. \tag{3}$$

This is in turn related to local Lipschitz continuity [2] around $\mathbf{x}$:

$$\text{SENS}_{\text{LIPS}}(\Phi, \mathbf{f}, \mathbf{x}, r) = \sup_{\|\delta\| \leqslant r} \frac{\|\Phi(\mathbf{x}) - \Phi(\mathbf{x} + \delta))\|}{\|\delta\|}, \tag{4}$$

Thus if an explanation has locally uniformly bounded gradients, it is locally Lipshitz continuous as well. In this paper, we consider a closely related measure, we term *max-sensitivity*, that measures the maximum change in the explanation with a small perturbation of the input $\mathbf{x}$.

**Definition 3.1.** Given a black-box function $\mathbf{f}$, explanation functional $\Phi$, and a given input neighborhood radius $r$, we define the max-sensitivity for explanation as:

$$\text{SENS}_{\text{MAX}}(\Phi, \mathbf{f}, \mathbf{x}, r) = \max_{\|\mathbf{y} - \mathbf{x}\| \leqslant r} \|\Phi(\mathbf{f}, \mathbf{y}) - \Phi(\mathbf{f}, \mathbf{x}))\|.$$

It can be readily seen that if an explanation is locally Lipshitz continuous, it has bounded max-sensitivity as well:

$$\text{SENS}_{\text{MAX}}(\Phi, \mathbf{f}, \mathbf{x}, r) := \max_{\|\delta\| \leqslant r} \|\Phi(\mathbf{f}, \mathbf{x} + \delta) - \Phi(\mathbf{f}, \mathbf{x}))\| \leqslant \text{SENS}_{\text{LIPS}}(\Phi, \mathbf{f}, \mathbf{x}, r)\, r, \tag{5}$$

The main attraction of the max-sensitivity measure is that it can be robustly estimated via Monte-Carlo sampling, as in our experiments. We point out that in certain cases, local Lipschitz continuity may be unbounded in a deep network (such as using ReLU activation function for gradient explanations, which is a common setting), but max-sensitivity is always finite given that explanation score is bounded, and thus is more robust to estimate. Can we then modify a given explanation so that it has lower sensitivity? If so, how much do we lower its sensitivity? There are two key objections to the very premise of these questions on how to lower sensitivity of an explanation. For the first objection, as we noted in the introduction, sensitivity provides only a partial measure of what is desired from an explanation. This can be seen from the fact that the optimal explanation that minimizes the above max-sensitivity measure is simply a constant explanation that just outputs a (potentially nonsensical) constant value for all possible test inputs. The second objection is that natural explanations might have a certain amount of sensitivity by their very nature, either because the model is sensitive, or because the explanations themselves are constructed by measuring the sensitivities of the predictor function, so that their sensitivities in turn is likely to be more than that of the function. In which case, we might not want to lower their sensitivities, since it might affect the fidelity of the explanation to the predictor function, and perhaps degrade the explanation towards the vacuous constant explanation.

As one key contribution of the paper, we show that it is indeed possible to reduce sensitivity "responsibly" by ensuring that it also *lowers the infidelity*, as we detail in the next section. We start by relating the sensitivity of an explanation to its infidelity, and then show that appropriately reducing the sensitivity can achieve two ends: lowering sensitivity of course, but surprisingly, also lowering the infidelity itself.

## 4  Reducing Sensitivity and Infidelity by Smoothing Explanations

In Section C in appendix, we show that if the explanation sensitivity is much larger than the function sensitivity around some input $\mathbf{x}$, the infidelity measure in turn will necessarily be large for some point around $\mathbf{x}$ (that is, loosely, infidelity is lower bounded by the difference in sensitivity of the explanation and the function). Given that a large class of explanations are based on sensitivity of the function at the test input, and such sensitivities in turn can be more sensitive to the input than the function itself, does that mean that sensitivity-based explanations are just fated to have a large infidelity? In this section, we show that this need not be the case: by appropriately lowering the sensitivity of any given explanation, we not only reduce its sensitivity, but also its infidelity.

Given any kernel $k(\mathbf{x}, \mathbf{z})$ over the input domain with respect to which we desire smoothness, and some explanation functional $\Phi(\mathbf{f}, \mathbf{z})$, we can define a smoothed explanation as $\Phi_k(\mathbf{f}, \mathbf{x}) := \int_{\mathbf{z}} \Phi(\mathbf{f}, \mathbf{z}) \, k(\mathbf{x}, \mathbf{z}) d\mathbf{z}$. When $k(\mathbf{x}, \mathbf{z})$ is set to the Gaussian kernel, $\Phi_k(\mathbf{f}, \mathbf{x})$ becomes Smooth-Grad [40]. We now show that the smoothed explanation is less sensitive than the original sensitivity averaged around $\mathbf{x}$.

**Theorem 4.1.** *Given a black-box function $\mathbf{f}$, explanation functional $\Phi$, the smoothed explanation functional $\Phi_k$,*
$$\mathrm{SENS}_{\mathrm{MAX}}(\Phi_k, \mathbf{f}, \mathbf{x}, r) \leqslant \int_{\mathbf{z}} \mathrm{SENS}_{\mathrm{MAX}}(\Phi, \mathbf{f}, \mathbf{z}, r) k(\mathbf{x}, \mathbf{z}) d\mathbf{z}.$$

Thus, when the sensitivity $\mathrm{SENS}_{\mathrm{MAX}}$ is large only along some directions $\mathbf{z}$, the averaged sensitivity could be much smaller than the worst case sensitivity over directions $\mathbf{z}$.

We now show that under certain assumptions, the infidelity of the smoothed explanation *actually decreases*. First, we introduce two relevant terms:

$$C_1 = \max_{\mathbf{x}} \frac{\int_{\mathbf{I}} \int_{\mathbf{z}} (\mathbf{f}(\mathbf{z}) - \mathbf{f}(\mathbf{z} - \mathbf{I}) - [\mathbf{f}(\mathbf{x}) - \mathbf{f}(\mathbf{x} - \mathbf{I})])^2 \, k(\mathbf{x}, \mathbf{z}) d\mathbf{z} d\mu_{\mathbf{I}}}{\int_{\mathbf{I}} \int_{\mathbf{z}} (\mathbf{I}^T \Phi(\mathbf{f}, \mathbf{z}) - [\mathbf{f}(\mathbf{x}) - \mathbf{f}(\mathbf{x} - \mathbf{I})])^2 \, k(\mathbf{x}, \mathbf{z}) d\mathbf{z} d\mu_{\mathbf{I}}}, \tag{6}$$

$$C_2 = \max_{\mathbf{x}} \frac{\int_{\mathbf{I}} \left( \int_{\mathbf{z}} \{\mathbf{I}^T \Phi(\mathbf{f}, \mathbf{z}) - [\mathbf{f}(\mathbf{x}) - \mathbf{f}(\mathbf{x} - \mathbf{I})]\} k(\mathbf{x}, \mathbf{z}) d\mathbf{z} \right)^2 d\mu_{\mathbf{I}}}{\int_{\mathbf{I}} \int_{\mathbf{z}} (\mathbf{I}^T \Phi(\mathbf{f}, \mathbf{z}) - [\mathbf{f}(\mathbf{x}) - \mathbf{f}(\mathbf{x} - \mathbf{I})])^2 \, k(\mathbf{x}, \mathbf{z}) d\mathbf{z} d\mu_{\mathbf{I}}}. \tag{7}$$

We note that when the sensitivity of $\mathbf{f}$ is much smaller than the sensitivity of $\mathbf{I}^T \Phi(\mathbf{f}, \cdot)$, the numerator of the term $C_1$ will be much smaller than the denominator of $C_1$, so that the term $C_1$ will be small. The term $C_2$ is smaller than one by Jensen's inequality, but in practice it may be much smaller than one when $\mathbf{I}^T \Phi(\mathbf{f}, \mathbf{z}) - [\mathbf{f}(\mathbf{x}) - \mathbf{f}(\mathbf{x} - \mathbf{I})]$ have different signs for varying $\mathbf{z}$. We now present our theorem which relates the infidelity of the smoothed explanation to that of the original explanation.

**Theorem 4.2.** *Given a black-box function $\mathbf{f}$, explanation functional $\Phi$, the smoothed explanation functional $\Phi_k$, some perturbation of interest $\mathbf{I}$, $C_1, C_2$ defined in (6) and (7) where $C_1 \leqslant \frac{1}{\sqrt{2}}$,*

$$INFD(\Phi_k, \mathbf{f}, \mathbf{x}) \leqslant \frac{C_2}{1 - 2\sqrt{C_1}} \int_{\mathbf{z}} INFD(\Phi, \mathbf{f}, \mathbf{z}) k(\mathbf{x}, \mathbf{z}) d\mathbf{z}.$$

When $\frac{C_2}{1 - 2\sqrt{C_1}} \leqslant 1$, we show that the infidelity of $\Phi_k$ is less than the infidelity of $\Phi$, as $\int_{\mathbf{z}} INFD(\Phi, \mathbf{f}, \mathbf{z}) k(\mathbf{x}, \mathbf{z}) d\mathbf{z}$ is usually very close to $INFD(\Phi, \mathbf{f}, \mathbf{z})$. This shows that smoothed explanation can be less sensitive and more faithful, which is validated in the experiments. Another direction to improve the explanation sensitivity and infidelity is to retrain the model, as we show in the appendix that adversarial traning leads to less sensitive and more faithful gradient explanations.

## 5  Experiments

**Setup.** We perform our experiments on randomly selected images from MNIST, CIFAR-10, and ImageNet. In our comparisons, we restrict local variants of the explanations to MNIST, since sensitivity of function values given pixel perturbations make more sense for grayscale rather than color images. To calculate our infidelity measure, we use the noisy baseline perturbation for local variants of the explanations, and the square removal for global variants of the explanations, and use Monte Carlo Sampling to estimate the measures. We use Grad, IG, GBP, and SHAP to denote

| Datasets | MNIST | |
|---|---|---|
| Methods | SENS$_{MAX}$ | INFD |
| Grad | 0.86 | 4.12 |
| Grad-SG | 0.23 | 1.84 |
| IG | 0.77 | 2.75 |
| IG-SG | 0.22 | 1.52 |
| GBP | 0.85 | 4.13 |
| GBP-SG | 0.23 | 1.84 |
| Noisy Baseline | 0.35 | 0.51 |

(a) Results for local explanations on MNIST dataset.

| Datasets | MNIST | | Cifar-10 | | Imagenet | |
|---|---|---|---|---|---|---|
| Methods | SENS$_{MAX}$ | INFD | SENS$_{MAX}$ | INFD | SENS$_{MAX}$ | INFD |
| Grad | 0.56 | 2.38 | 1.15 | 15.99 | 1.16 | 0.25 |
| Grad-SG | 0.28 | 1.89 | 1.15 | 13.94 | 0.59 | 0.24 |
| IG | 0.47 | 1.88 | 1.08 | 16.03 | 0.93 | 0.24 |
| IG-SG | 0.26 | 1.72 | 0.90 | 15.90 | 0.48 | 0.23 |
| GBP | 0.58 | 2.38 | 1.18 | 15.99 | 1.09 | 0.15 |
| GBP-SG | 0.29 | 1.88 | 1.15 | 13.93 | 0.41 | 0.15 |
| SHAP | 0.35 | 1.20 | 0.93 | 5.78 | – | – |
| Square | 0.24 | 0.46 | 0.99 | 2.27 | 1.33 | 0.04 |

(b) Results for global explanations on MNIST, Cifar-10 and imagenet.

Table 1: Sensitivity and Infidelity for local and global explanations.

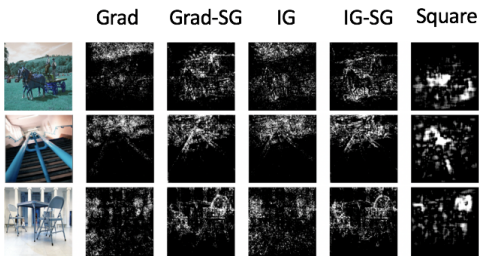

Figure 1: Examples of explanations on Imagenet.

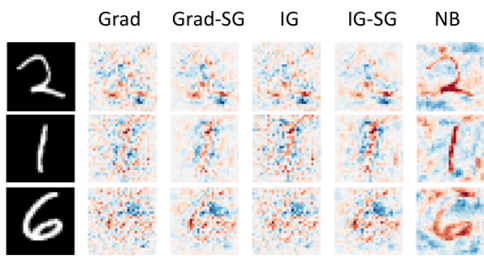

Figure 2: Examples of local explanations on MNIST.

vanilla gradient [37], integrated gradient [43], Guided Back-Propagation [41], and KernelSHAP [25] respectively, and add the postfix "-SG" when Smooth-Grad [40] is applied. We call the optimal explanation with respect to the perturbation Noisy Baseline and Square Removal as NB and Square for simplicity. We provide more exhaustive details of the experiments in the appendix.

**Explanation Sensitivity and Infidelity.** We show results comparing sensitivity and infidelity for local explanations on MNIST and global explanations on MNIST, CIFAR-10, and ImageNet in Table 1. Recalling the discussion from Section 2.5, global explanations include a point-wise multiplication with the image minus baseline, but local explanations do not. We observe that the noisy baseline and square removal optimal explanations achieve the lowest infidelity, which is as expected, since they explicitly optimize the corresponding infidelity. We also observe that Smooth-Grad improves *both* sensitivity and infidelity for all base explanations across all datasets, which corroborates the analysis in section 4, and also addresses plausible criticisms of lowering sensitivity via smoothing: while one might expect such smoothing to increase infidelity, modest smoothing actually improves infidelity. We also perform a sanity check experiment when the perturbation follows that in SHAP (Defined in Prop.2.5), and we verify that SHAP has the lowest infidelity for this perturbation.

In the Appendix, we investigate how varying the smoothing radius for Smooth-Grad impacts the sensitivity and infidelity. We also provide an analysis of how adversarial training of robust networks can also lower both sensitivity and infidelity (which is useful in the case where we can retrain the model), which we validate both measures are lowered in additional experiments.

**Visualization.** For a qualitative evaluation, we show several examples of global explanations on ImageNet, and local explanations on MNIST. The explanations optimizing our infidelity measure with respect to Square and Noisy Baseline (NB) perturbations, show a cleaner saliency map, highlighting the actual object being classified, when compared to then other explanations. For example, Square is the only explanation that highlights the whole bannister in the second image of Figure 1. For local examples on MNIST, NB clearly shows the digits, as well as regions that would increase the prediction score if brightened, such as the region on top of the number 6, which gives more insight into the behavior of the model. We also observe that SG provides a cleaner set of explanations, which validates the experimental results in [40], as well as our analysis in Section 4. We provide a more complete set of visualization results with higher resolution in the appendix.

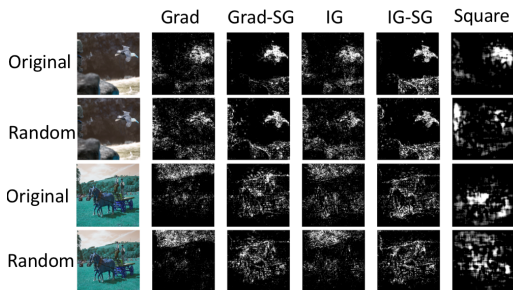
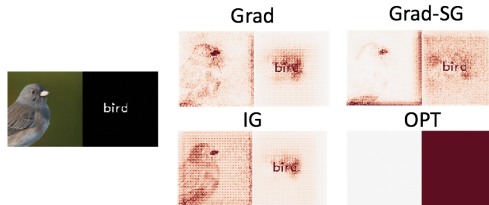

Figure 3: Examples of various explanations for the original model and the randomized model. More in appendix.

Figure 4: One example of explanations where the approximated ground truth is the right block (model focuses on the text). Some explanations focus on both text and image, so that just from these explanations, might be difficult to infer the ground truth feature used. More examples in appendix.

**Human Evaluation.** We perform a controlled experiment to validate whether the infidelity measure aligns with human intuitions in a setting where we have an approximated ground truth feature for our model, following the setting of [18]. We create a dataset of two classes (bird and frog), with the image of the bird or frog in one half of the overall image, and just the caption in the other half (as shown in Figure 4). The images are potentially noisy with noise probability $p \in \{0, 0.6\}$: when $p = 0$, the image always agrees with the caption, and when $p = 0.6$, we randomize the image 60 percent of the time to a random image of another class. We train two models which both achieve testing accuracy above 0.95, where one model only relies on the image and the other only relies on the caption[7]. We then show the original input with aligned image and text, the prediction result, along with the corresponding explanations of the model (among Grad, IG, Grad-SG, and OPT) to humans, and test how often humans are able to infer the approximated ground truth feature (image or caption) the model relies on. The optimal explanation (OPT) is the explanation that minimizes our infidelity measure with respect to perturbation **I** defined as the right half or the left half of the image (since the location of the caption is in one half of the overall image in our case; but note that in more general settings, we could simply use a caption bounding box detector to specify our perturbations). Our human study includes 2 models, 4 explanations, and 16 test users, where each test user did a series of 8 tasks (2 models × 4 explanations) on random images. We report the average human accuracy and the infidelity measure for each explanation models in Table 3. We observe that unsurprisingly OPT has the best infidelity score by construction, and we also observe that the infidelity aligns with human evaluation result in general. This suggests that a faithful explanation communicates the important feature better in this setting, which validates the usefulness of the objective measure.

**Sanity Check.** Recent work in the interpretable machine learning literature [12, 1] has strongly argued for the importance of performing sanity checks on whether the explanation is at least loosely related to the model. Here, we conduct the sanity check proposed by Adebayo et al. [1], to check if explanations look different when the network being explained is randomly perturbed. One might expect that explanations that minimize infidelity will naturally be faithful to the model, and consequently pass this sanity check. We show visualizations for various explanations (with and without absolute values) of predictions by a pretrained Resnet-50 model and a randomized Resnet-50 model where the final fully connected layer is randomized in Figure 3. We also report the average rank correlation of the explanations for the original model and the randomized model in Table 2. All explanations without the absolute value pass the sanity check, but the rank correlation for explanations with the absolute value between the original model and the randomized model is high. In this case, Square has the lowest rank correlation and the visualizations for two models look the most distinct, which supports the hypothesis that an explanation with low infidelity is also faithful to the model. More examples are included in the appendix.

## 6 Related Work

Our work focuses on placing attribution-based explanations on an objective footing. We begin with a brief and necessarily incomplete review of recent explanation mechanisms, and then discuss recent approaches to place these on an objective footing. While attribution-based explanations are the most popular form of explanations, other types of explanations do exist. Sample-based explanation

|       | Grad | Grad-SG | IG   | IG-SG | Square |
|-------|------|---------|------|-------|--------|
| Corr  | 0.17 | 0.10    | 0.18 | 0.16  | 0.13   |
| Corr (abs) | 0.57 | 0.62 | 0.61 | 0.62  | 0.28   |

|       | Grad | Grad-SG | IG   | OPT  |
|-------|------|---------|------|------|
| Infid. | 0.55 | 0.38   | 0.35 | 0.00 |
| Acc.  | 0.47 | 0.50    | 0.53 | 0.88 |

Table 2: Correlation of the explanation between the original model randomized model for the sanity check.

Table 3: The infidelity and the accuracy human are able to predict the input blocked used based on the explanations.

methods attribute the decision of the model to previously observed samples [21, 45, 17]. Concept-based explanation methods seek to explain the decision of the model by high-level human concepts [18, 14, 6]. However, attribution-based explanations have the advantage that they are generally applicable to a wide range of tasks and they are easy to understand. Among attribution-based explanations, perturbation-based attributions measure the prediction difference after perturbing a set of features. Zeiler & Fergus [47] use such perturbations with grey patch occlusions on CNNs. This was further improved by [49, 7] by including a generative model, similar in spirit to counterfactual visual explanations [15]. Gradient based attribution explanations [5, 38, 47, 41, 35] range from explicit gradients, to variants that leverage back-propagation to address some caveats with simple gradients. As shown in [3], many recent explanations such as $\epsilon$-LRP [4], Deep LIFT [37], and Integrated Gradients [43] can be seen as variants of gradient explanations. There are also approaches that average feature importance weights by varying the active subsets of the set of input features (e.g. over the power set of the set of all features), which has roots in cooperative game theory and revenue division [11, 25].

Among works that place these explanations on a more objective footing are those that focus on improving the sensitivity of explanations. To reduce the noise in gradient saliency maps, Kindermans et al. [19] propose to calculate the signal in the image by removing distractors. SmoothGrad [40] generating noisy images via additive Gaussian noise and average the gradient of the sampled images. Another form of sensitivity analysis proposed by Ribeiro et al. [32] approximates the behavior of a complex model by an locally linear interpretable model, which has been extended by [46, 30] in different domains. The reliability of these attribution explanations is a key problem of interest. Adebayo et al. [1] has shown that several saliency methods are insensitive to random perturbations in the parameter space, generating the same saliency maps even when the parameter space is randomized. Ghorbani et al. [13], Zhang et al. [48] shows that it is possible to generate a perceptively indistinguishable image that changes the saliency explanations significantly. In this work, we show that the optimal explanation that optimizes fidelity passes the sanity check in [1] and smoothing explanations with SmoothGrad [40] lowers the sensitivity and infidelity of explanations, which sheds light on how to generate more robust explanations that does not degrade the fidelity, which addresses the concerns for saliency explanations. There are also works that proposes objective evaluations for saliency explanations. Montavon et al. [28] use explanation continuity as an objective measure of explanations, and observe that discontinuities may occur for gradient-based explanations, while variants such as deep Taylor LRP [4] can achieve continuous explanations, as compared to simple gradient explanations. Samek et al. [34] evaluate the explanations by the area over perturbation curve while removing the most salient features. Dabkowski & Gal [10] uses object localisation metrics to evaluate the closeness of the saliency and the actual object. Kindermans et al. [20] posit that a good explanations should fulfill input invariance. Hooker et al. [16] propose to remove salient features and retrain the model to evaluate explanations.

## 7 Conclusion

We propose two *objective* evaluation metrics, naturally termed infidelity and sensitivity, for machine learning explanations. One of our key contributions is to show that a large number of existing explanations can be unified, as they all optimize the infidelity with respect to various perturbations. We then show that the explanation that optimizes the infidelity can be seen as a combination of two existing explanation methods with a kernel with respect to the perturbation. We then propose two perturbations and their respective optimal explanations as new explanations. Another key contribution of the paper is that we show both theoretically and empirically that there need not exist a trade-off between sensitivity and infidelity, as we may improve the sensitivity as well as the infidelity of explanations by the right amount of smoothing. Finally, we validate that our infidelity measurement aligns with human evaluation in a setting where the ground truth of explanations is given.

**Acknowlegement**   We acknowledge the support of DARPA via FA87501720152, and Accenture.

## Footnotes

[6]Implementation available at `https://github.com/chihkuanyeh/saliency_evaluation`.

[7]When $p = 0$, the trained model solely relies on the image (accuracy for image only input is 0.9, but accuracy for caption only input is 0.5). When $p = 0.6$, the trained model only relies on the caption (the accuracy for caption only input is 0.98 but the accuracy for image only input is 0.5

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
