[Supplementary Material]

# A   A Case study of infidelity and sensitivity for gradient explanations

Gradient is usually preferred as an explanation due to the low computational complexity, and thus it is interesting to measure how sensitive and faithful is gradient. In this section, we provide an upper bound of the sensitivity of the gradient explanation for generic deep neural networks as well as justifications on how adversarial training may lead to explanations that are less sensitive and more faithful.

## A.1   Sensitivity bound for gradient in Neural Networks

We first consider the sensitivity when the explanations themselves are gradients $\Phi_{\mathbf{f}} = \nabla_{\mathbf{x}}\mathbf{f}$ of the machine-learnt predictor function $\mathbf{f}$. As a concrete example, consider deep neural networks with SoftPlus activations, which are a differentiable close approximation of the commonly used ReLU activation.

**Proposition A.1.** *Suppose the predictor $\mathbf{f}$ is a $l$-layer Softplus neural network with weights $W_i$ at layer $i$, and bias at each layer equal to $0$, so that $\mathbf{f}(\mathbf{x}) = \sigma(W_l\sigma(W_{l-1}...\sigma(W_1\mathbf{x})))$, where $\sigma(v) = \log(1 + \exp(v))$. Let $\Phi(\mathbf{f}, \mathbf{x})$ denote the gradient explanation at point $\mathbf{x}$, so that $\Phi(\mathbf{f}, \mathbf{x}) = \nabla_{\mathbf{x}}\mathbf{f}(\mathbf{x})$. Then the sensitivity of $\Phi(\mathbf{f}, \mathbf{x})$ is upper bounded as: $\mathrm{SENS}_{\mathrm{MAX}} \leqslant \prod_{i=1}^{l} \frac{||W_i||^2}{4}r$, under $\ell_2$ distance for the distance metrics $D$ and $\rho$.*

The proposition follows naturally by observing that the Lipschitz constant of $\nabla_{\mathbf{x}}\mathbf{f}$ is upper bounded by $\prod_{i=1}^{l} ||W_i||^2 L(\sigma) \leqslant \prod_{i=1}^{l} \frac{||W_i||^2}{4}$ with respect to $\ell_2$ distance. Consequently, (5) holds for $L = \prod_{i=1}^{l} \frac{||W_i||^2}{4}$ and $\Phi(\mathbf{f}, \mathbf{x}) = \nabla_{\mathbf{x}}\mathbf{f}(\mathbf{x})$.

## A.2   Infidelity and Sensitivity upper bound for Gradient

For a function with a bounded Hessian around input $\mathbf{x}$, such that $\sup_{\|\delta\| \leqslant \sup(\|I\|)} \|\nabla_{\mathbf{x}}^2\mathbf{f}(\mathbf{x} + \delta)\| \leqslant L$, we may upper bound the infidelity score for the gradient explanation.

$$\min_{\Phi(\mathbf{f},\mathbf{x})} \mathbb{E}_{\mathbf{I}}[||\mathbf{I}^T\Phi(\mathbf{f}, \mathbf{x}) - (\mathbf{f}(\mathbf{x}) - \mathbf{f}(\mathbf{x} - \mathbf{I}))||^2]$$

$$\leqslant \mathbb{E}_{\mathbf{I}}[||\mathbf{I}^T||^2||\nabla_{\mathbf{x}}\mathbf{f}(\mathbf{x}) - \int_0^1 \nabla_{\mathbf{x}}\mathbf{f}(\mathbf{x} - t\mathbf{I})dt||^2]$$

$$= \mathbb{E}_{\mathbf{I}}[||\mathbf{I}^T||^2||\int_0^1 (\nabla_{\mathbf{x}}\mathbf{f}(\mathbf{x}) - \nabla_{\mathbf{x}}\mathbf{f}(\mathbf{x} - t\mathbf{I}))dt||^2]$$

$$= \mathbb{E}_{\mathbf{I}}[||\mathbf{I}^T||^2||\int_0^1 (t\mathbf{I}^T\int_0^1 \nabla_{\mathbf{x}}^2\mathbf{f}(\mathbf{x} - st\mathbf{I})ds)dt||^2]$$

$$\leqslant \mathbb{E}_{\mathbf{I}}[||\mathbf{I}^T||^4||\int_0^1 (t\int_0^1 \nabla_{\mathbf{x}}^2\mathbf{f}(\mathbf{x} - st\mathbf{I})ds)dt||^2]$$

$$\leqslant \mathbb{E}_{\mathbf{I}}[||\mathbf{I}^T||^4] \cdot \frac{L^2}{2}$$

Intuitively, gradient is the completely faithful explanation to a model with no curvature at all, and thus a model with a lower Hessian upper bound may lead to a better gradient infidelity score. We further show the sensitivity of gradient explanations can also be bounded by the hessian upper bound when $r \leqslant \sup(\|I\|)$,

$$\max_{\|\delta\| \leqslant r} \|\nabla_{\mathbf{x}}\mathbf{f}(\mathbf{x} + \delta) - \nabla_{\mathbf{x}}\mathbf{f}(\mathbf{x})\| = \max_{\|\delta\| \leqslant r} \frac{\|\nabla_{\mathbf{x}}\mathbf{f}(\mathbf{x} + \delta) - \nabla_{\mathbf{x}}\mathbf{f}(\mathbf{x})\|}{\|\delta\|}\|\delta\|$$

$$\leqslant \max_{\|\delta\| \leqslant r} \|\nabla_{\mathbf{x}}^2\mathbf{f}(\mathbf{x} + \delta)\|\|\delta\|$$

$$\leqslant L\,r,$$

which shows that a lower Hessian upper bound leads to improved infidelity and sensitivity for gradient explanations.

## A.3 Optimizing Explanation Infidelity and Gradient Sensitivity by Adversarial Training

In this section, we explore a different approach to lower the sensitivity of explanations and improve the infidelity score by having the freedom to retrain the model, which has been explored in Lee et al. [23], Ross & Doshi-Velez [33]. We propose to directly optimize the Hessian upper bound $\sup_{\|\delta\| \leqslant r} \|\nabla_{\mathbf{x}}^2 \Phi(\mathbf{x} + \delta)\|$, which we showed in A.2 this leads to better infidelity and sensitivity for the gradient explanations.

One direct way is to lower the Hessian of models is to impose a Hessian regularizer during training. However, this may be infeasible as this requires the calculation of the gradient for Hessian, which may be expensive. Another way may be to impose L2 weight regularizer on the network, as by Corollary A.1 this leads to lower sensitivity of the gradient, which is related to the Hessian.

An alternative way to robustify gradient based explanations is to learn a model with smooth gradients. We show that models learned through "adversarial training" have smooth gradients and as a result the gradient based explanations of these models are naturally robust to perturbations. An adversarial perturbation at a point $\mathbf{x}$ with label $y$, for any classifier $\mathbf{f}$ is defined as any perturbation $\delta, \|\delta\| \leqslant \epsilon$ such that $\mathbf{f}(\mathbf{x} + \delta) \neq y$. The *adversarial loss* at a point $x$ is defined as: $\ell_{adv}(\mathbf{f}, \mathbf{x}, y) = \sup_{\delta:\|\delta\| \leqslant \epsilon} \ell(\mathbf{f}(\mathbf{x} + \delta), y)$, where $\ell$ is a classification loss such as logistic loss. The expected adversarial risk of a classifier $\mathbf{f}$ is then defined as: $\mathbb{E}\left[\ell_{adv}(\mathbf{f}, \mathbf{x}, y)\right]$. The goal in adversarial training is to minimize the expected adversarial risk. We now show that minimizing expected adversarial risk results in models with smooth gradients.

**Theorem A.1.** *Consider the binary classification setting, where $y \in \{0, 1\}$ and $\ell$ is the logistic loss. Suppose $\mathbf{f}$ is twice differentiable w.r.t $\mathbf{x}$. For any $\epsilon > 0$, the adversarial training objective can be upper bounded as*

$$
\begin{aligned}
\mathbb{E}\left[\ell_{adv}(\mathbf{f}(\mathbf{x} + \delta), y)\right] \leqslant & \mathbb{E}\left[\ell(\mathbf{f}(\mathbf{x}), y)\right] + \epsilon \mathbb{E}\left[\|\nabla_{\mathbf{x}}\mathbf{f}(\mathbf{x})\|_*\right] \\
& + \frac{\epsilon^2}{2}\mathbb{E}\left[\sup_{\|\delta\| \leqslant \epsilon} \|\nabla_{\mathbf{x}}^2 \mathbf{f}(\mathbf{x} + \delta)\|\right],
\end{aligned} \tag{8}
$$

*where $\|.\|_*$ is the dual norm of $\|.\|$, which is defined as $\|z\|_* = \sup_{\|x\| \leqslant 1} x^T z$.*

Notice the two terms in the upper bound which penalize the norm of the gradient and Hessian. It can be seen that by optimizing the adversarial risk, we are effectively optimizing a gradient and hessian norm penalized risk. This suggests that optimizing the adversarial risk can lead to classifiers with small and "smooth" gradients, which are naturally more robust to perturbations.

A number of techniques have been proposed for minimization of the expected adversarial risk [26, 31, 39, 44]. In our experiments, we use the Projected Gradient Descent (PGD) technique of [26] to train an adversarially robust network. We conclude the section with a discussion on other potential approaches one could take to obtain models with robust gradients. One natural technique is to add a regularizer which penalizes gradients with large norms; that is, add a gradient norm penalty to the training objective. Ross & Doshi-Velez [33] study this approach and empirically show that this results in more explainable models. However, a drawback of this approach is that it has a large training time, since it has to deal with Hessians during training.

Another alternative is to inference the model by averaging the results while adding some Gaussian noise to the input, the gradient of the model correspond to that of SG, which we show to be more robust. Interestingly, recent research has shown that such random ensemble also lead to more robust prediction [24, 9]. The main advantage of using adversarial training over gradient regularization is that adversarial robustness is an active research area and a number of efficient techniques are being designed for faster training.

## A.4 Proof of Theorem A.1

We consider logistic loss, a convex surrogate of the 0/1 loss, which is defined as

$$
\ell(\mathbf{f}(\mathbf{x}), y) = -\log \frac{e^{y\mathbf{f}(\mathbf{x})}}{1 + e^{\mathbf{f}(\mathbf{x})}}.
$$

We now try to show that minimizing adversarial risk results in classifiers with smooth gradients. First note that $\mathbf{f}(\mathbf{x} + \delta)$ can be written as

$$\mathbf{f}(\mathbf{x} + \delta) = \mathbf{f}(\mathbf{x}) + \int_{t=0}^{1} \nabla \mathbf{f}(\mathbf{x} + t\delta)^T \delta \; dt.$$

We also have

$$\nabla \mathbf{f}(\mathbf{x} + t\delta) = \nabla \mathbf{f}(\mathbf{x}) + \int_{s=0}^{t} \nabla^2 \mathbf{f}(\mathbf{x} + s\delta)\delta \; ds.$$

Substituting this in the previous expression gives us

$$\mathbf{f}(\mathbf{x} + \delta) = \mathbf{f}(\mathbf{x}) + \nabla \mathbf{f}(\mathbf{x})^T \delta + \int_{t=0}^{1}\int_{s=0}^{t} \delta^T \nabla^2 \mathbf{f}(\mathbf{x} + s\delta)\delta \; dsdt.$$

This can be upper bounded as follows

$$\mathbf{f}(\mathbf{x} + \delta) \leqslant \mathbf{f}(\mathbf{x}) + \epsilon \|\nabla \mathbf{f}(\mathbf{x})\|_* + \frac{\epsilon^2}{2} \sup_{\|\delta\| \leqslant \epsilon} \|\nabla^2 \mathbf{f}(\mathbf{x} + \delta)\|,$$

where $\|.\|_*$ is the dual norm of $\|.\|$.

Let $u(\mathbf{x})$ be defined as

$$u(\mathbf{x}) = \epsilon \|\nabla \mathbf{f}(\mathbf{x})\|_* + \frac{\epsilon^2}{2} \sup_{\|\delta\| \leqslant \epsilon} \|\nabla^2 \mathbf{f}(\mathbf{x} + \delta)\|.$$

Some algebra shows that $\ell(\mathbf{f}(\mathbf{x} + \delta), y)$ can be upper bounded by

$$\ell(\mathbf{f}(\mathbf{x} + \delta), y) \leqslant \ell(\mathbf{f}(x) + (1 - 2y)u(\mathbf{x}), y) \leqslant \ell(\mathbf{f}(\mathbf{x}), y) + u(\mathbf{x}).$$

So we have the following upper bound for our objective

$$\mathbb{E}\left[\sup_{\delta:\|\delta\| \leqslant \epsilon} \ell(\mathbf{f}(\mathbf{x} + \delta), y)\right] \leqslant \mathbb{E}\left[\ell(\mathbf{f}(\mathbf{x}), y)\right]$$

$$+ \underbrace{\epsilon \mathbb{E}\left[\|\nabla \mathbf{f}(\mathbf{x})\|_*\right] + \frac{\epsilon^2}{2} \mathbb{E}\left[\sup_{\|\delta\| \leqslant \epsilon} \|\nabla^2 \mathbf{f}(\mathbf{x} + \delta)\|\right]}_{\text{Regularization Term}}. \qquad (9)$$

## A.5   A toy example

We now consider a simple toy example and use it to illustrate why SmoothGrad might result in more faithful explanations. Consider the following function in $2d$ Euclidean space: $f(a, b) = \max(a, b) - \lfloor|a - b|\rfloor/2$ if $\lfloor|a - b|\rfloor \mod 2 \equiv 0$, and $f(a, b) = \min(a, b) + (\lfloor|a - b|\rfloor + 1)/2$ if $\lfloor|a - b|\rfloor \mod 2 \equiv 1$. This function can be easily shown to be continuous in $a, b$. The gradient of $f$ is given by: $\nabla_{\mathbf{x}} f(a, b) = (1, 0)$ if $\lfloor|a - b|\rfloor \mod 2 \equiv 0$, and $\nabla_{\mathbf{x}} f(a, b) = (0, 1)$ if $\lfloor|a - b|\rfloor \mod 2 \equiv 1$. We visualize the gradient in Figure 5. It can be seen that the gradient is very sensitive with respect to the input. The point $(20, 11.9)$, which has a function value of 16, has a gradient $(1, 0)$. Whereas, the perturbed point $(20, 12.1)$, which is close to $(20, 11.9)$, and has a function value of 16.1, has a very different gradient of $(0, 1)$. Apart from being sensitive, the gradient is also unfaithful to the function output. The gradient $(1, 0)$ at $(20, 11.9)$ implies that only feature $a$ is relevant to the function value. However, if we increase the value of feature $b$ to 15.9, the function value increases to 18. Therefore, the gradient explanation clearly does not reflect the fact that feature $b$ is relevant to the function output. Here, gradient-SG is close to $(0.5, 0.5)$ (computed with the kernel to be uniform over an enormous ball), which is more faithful to the function output and less sensitive (it is close to $(0.5, 0.5)$ for all inputs). This toy example provides insights on how SmoothGrad may achieve more faithful explanations.

# B   Additional Experiments

## B.1   Detailed Experiment Settings

For MNIST, we train our own CNN model with testing accuracy above 99 percent. For cifar-10, we use a baseline wide-resnet model with 94 percent testing accuracy. In our experiments we compare

Figure 5: Visualization of the gradient and function value as we perturb the input point for a toy example. The blue region has the gradient of $(1, 0)$, while the red region has the gradient of $(0,1)$.

simple gradients (Grad), integrated gradients (IG), Guided Back-Propagation (GBP), and the Smooth-Grad (SG) technique applied on all above explanations. To compute the sensitivity scores defined in Definitions 3.1, we randomly sample 50 points with Monte-Carlo sampling. Following the adversarial literature, we set the norm for $\|y - x\|$ in Definition 3.1 and the norm for $\|\mathbf{u}\|$ in definition C.1 to be $L_\infty$ norm and the value of the maximum perturbation $r$ is set to $0.1$ for all datasets. To allow fair comparisons among different explanation methods, we normalize the explanation to have unit norm before calculating the sensitivity, and perform optimal scaling before calculating the infidelity. For the Smooth-Grad smoothing kernel, we set it to be an uniform kernel for a $L_\infty$ ball around the data point with the radius as a parameter, which is set to $0.2$ for all datasets, and we choose 200 points to perform Monte Carlo Sampling for Smooth-Grad. We call the optimal explanation with respect to the perturbation Noisy Baseline and Square Removal as NB and Square for simplicity, and the optimal solution is calculated via Monte Carlo Sampling of 20,000 points. For calculating infidelity and sensitivity, 1000 points are sampled to evaluate the infidelity, and 50 points are sampled to evaluate the sensitivity. We report the average sensitivity and infidelity over 50 instances.

The baseline image is set to the 0 for all explanations. We add a small identity matrix to $\int \mathbf{II}^T d\mu_{\mathbf{I}}$ when $\int \mathbf{II}^T d\mu_{\mathbf{I}}$ is not invertible (or take the Pseudo-inverse), which makes the optimal explanation much more stable. For Imagenet, we consider a $2 \times 2$ superpixel scheme to lower the memory usage for calculation of the inverse matrix for the Square optimal explanation. For SHAP, we adopt the KernelSHAP version introduced in [25], but the SHAP kernel regression does not scale well to high dimension in imagenet data and only produces random noises even with the $2 \times 2$ superpixel, as future works have been developed to scale SHAP to high dimensions more efficiently [8]. However, they are not scaled to such high dimensions as imagenet images, and thus we do not report numbers for SHAP on imagenet. For Square perturbation, we limit to square size from $1 \times 1$ to $10 \times 10$ in MNIST, $1 \times 1$ to $15 \times 15$ in Cifar-10, and $10 \times 10$ to $30 \times 30$ in imagenet.

### B.2 Implementation Tricks

When implementing SHAP and Square, where we have a mapping $h_{\mathbf{x}}$ that maps a simplified binary inputs $\mathbf{z}$ to real valued inputs, where the perturbation can be written as $\mathbf{I} = h_{\mathbf{x}}(\mathbf{z})$. We observe that $\int \mathbf{II}^T d\mu_{\mathbf{I}}$ is not invertible when the original input $\mathbf{x}$ contains some 0 features, which is always the case in the MNIST dataset with black background. While there are several workarounds such as pseudo-inverse or adding a small identity matrix to make the matrix invertible, we derive an alternative form for the optimal solution with the multiplication to the image as:

$$\Phi^*(\mathbf{f}, \mathbf{x}) \odot \mathbf{x} = \left( \int \mathbf{zz}^T d\mu_{\mathbf{I}} \right)^{-1} \left( \int \mathbf{zI}^T \mathrm{IG}(\mathbf{f}, x, \mathbf{I}) d\mu_{\mathbf{I}} \right),$$

which has a simple proof shown below:

$$\Phi^*(\mathbf{f}, \mathbf{x}) = \arg \min_{\Phi(\mathbf{f},\mathbf{x})} \mathbb{E}_{\mathbf{I}}[\|\mathbf{I}^T \Phi(\mathbf{f}, \mathbf{x}) - (\mathbf{f}(\mathbf{x}) - \mathbf{f}(\mathbf{x} - \mathbf{I}))\|^2]$$
$$= \arg \min_{\Phi(\mathbf{f},\mathbf{x})} \mathbb{E}_{\mathbf{I}}[\|(\mathbf{z} \odot \mathbf{x})^T \Phi(\mathbf{f}, \mathbf{x}) - (\mathbf{f}(\mathbf{x}) - \mathbf{f}(\mathbf{x} - \mathbf{I}))\|^2] \qquad (10)$$
$$= \arg \min_{\Phi(\mathbf{f},\mathbf{x})} \mathbb{E}_{\mathbf{I}}[\|\mathbf{z}^T (\mathbf{x} \odot \Phi(\mathbf{f}, \mathbf{x})) - (\mathbf{f}(\mathbf{x}) - \mathbf{f}(\mathbf{x} - \mathbf{I}))\|^2],$$

whose analytical solution is

$$\left( \int \mathbf{zz}^T d\mu_{\mathbf{I}} \right)^{-1} \left( \int \mathbf{zI}^T \mathrm{IG}(\mathbf{f}, x, \mathbf{I}) d\mu_{\mathbf{I}} \right).$$

| Datasets | MNIST | | MNIST-Robust | |
|---|---|---|---|---|
| Methods | SENS$_{MAX}$ | INFD | SENS$_{MAX}$ | INFD |
| Grad | 2.32 | 2.42 | 0.21 | 0.36 |
| Grad-SG | 1.82 | 1.88 | 0.13 | 0.35 |
| IG | 2.05 | 1.78 | 0.16 | 0.21 |
| IG-SG | 1.69 | 1.77 | 0.11 | 0.2 |
| GBP | 2.36 | 2.42 | 0.21 | 0.36 |
| GBP-SG | 1.83 | 1.82 | 0.13 | 0.35 |
| SHAP | 0.35 | 1.20 | 0.23 | 0.14 |
| Square | 0.24 | 0.46 | 0.11 | 0.06 |

Table 4: Sensitivity and Infidelity for global explantion in MNIST for robust and regular network.

Therefore, by carefully selecting $\mathbf{z}$, $\left(\int \mathbf{z}\mathbf{z}^T d\mu_{\mathbf{I}}\right)$ can be invertible, and thus we use this form whenever we are calculating SHAP and Square. We also use the form:

$$\mathbb{E}_{\mathbf{I}}[\|\mathbf{z}^T(\mathbf{x} \odot \Phi(\mathbf{f}, \mathbf{x})) - (\mathbf{f}(\mathbf{x}) - \mathbf{f}(\mathbf{x} - \mathbf{I}))\|^2]$$

to evaluate the infidelity of explanations under square perturbations, that is, we set the perturbations to binary and evaluate the infidelity for global explanations, which is equal to the original form.

## B.3 Sensitivity and Infidelity for Varying Smoothing Strength

We further investigate how the infidelity and sensitivity varies for increasing smoothing radius for Smooth-Grad. We show the infidelity and sensitivity for Smooth-Grad when the smoothing radius for Smooth-Grad is increased from 0.1 to 2.0 on the MNIST dataset in Figure 6. We observe that the infidelity first decreases as the smoothing radius increases, but then increases gradually. On the other hand, the sensitivity of the smoothed explanation decreases while the smoothing radius increases. The experiment shows that although the least sensitive explanation are not faithful, we can improve the infidelity and sensitivity simultaneously with the right amount of smoothing. All saliency maps is normalized to zero mean and unit variance before visualization.

Figure 6: Infidelity and Sensitivity for local Grad-SG for increasing smoothing radius on MNIST.

## B.4 Infidelity and Sensitivity for Robust Network

As shown in section A.3, an adversarially trained network leads to a lower sensitivity and infidelity for the gradient explanation. We therefore report the sensitivity and infidelity of various explanations on a regularly trained MNIST and a adversarially trained MNIST in Table 4. We adopt the model trained in [26]. We observe the robust model yields lower sensitivity and infidelity for all explanations.

## B.5 Additional Viusalization Examples

We show additional visualization results in Figure 8, 7, and 9.

# C   A Connection between Explanation and Model Sensitivity and Infidelity

We then introduce a robust version of explanation fidelity which measures the maximum infidelity while adding a small perturbation to $\mathbf{x}$:

**Definition C.1.** Given a black-box function $\mathbf{f}$, explanation functional $\Phi$, a random variable $\mathbf{I}$ which represents meaningful perturbation of interest, we define the robust fidelity of $\mathbf{x}$ as:

$$\text{RINFD}(\Phi, \mathbf{f}, \mathbf{x}) = \max_{\|\mathbf{u}\| \leqslant r} \text{INFD}(\Phi, \mathbf{f}, \mathbf{x} + \mathbf{u})$$

$$= \max_{\|\mathbf{u}\| \leqslant r} \mathbb{E}_{\mathbf{I}}[\|\mathbf{I}^T \Phi(\mathbf{f}, \mathbf{x} + \mathbf{u}) - (\mathbf{f}(\mathbf{x} + \mathbf{u}) - \mathbf{f}(\mathbf{x} + \mathbf{u} - \mathbf{I}))\|^2].$$

We note that the optimal explanation that optimizes the robust infidelity equals to the optimal explanation that optimizes the infidelity. Therefore, by introducing the robust infidelity, we do not modify the optimal explanation but only capture a more robust measurement of the infidelity score. We introduce the following theorem that gives a lower bound for the robust infidelity that relates to the explanation sensitivity. The intuition is that by Definition C.1, $\mathbf{I}^T \Phi(\mathbf{f}, \mathbf{x} + \mathbf{u})$ and $\mathbf{f}(\mathbf{x}+\mathbf{u})-\mathbf{f}(\mathbf{x}+\mathbf{u}-\mathbf{I})$ should be close for all $\mathbf{u}$. However, if $\mathbf{I}^T\Phi(\mathbf{f}, \mathbf{x}+\mathbf{u})$ and $\mathbf{f}(\mathbf{x}+\mathbf{u})-\mathbf{f}(\mathbf{x}+\mathbf{u}-\mathbf{I})$ have a very different sensitivity when perturbing $\mathbf{u}$, the difference will naturally be large for some $\mathbf{u}$.

**Theorem C.1.** *Given a black-box function $\mathbf{f}$, explanation functional $\Phi$, a random variable $\mathbf{I}$, and let $A(\mathbf{x}) = \max_{\|\mathbf{u}\| \leqslant r} \mathbb{E}_{\mathbf{I}}[\|\mathbf{I}^T\Phi(\mathbf{f}, \mathbf{x}+\mathbf{u})-\mathbf{I}^T\Phi(\mathbf{f}, \mathbf{x})\|]$, and $B(\mathbf{x}) = \max_{\|\mathbf{u}\| \leqslant r} \mathbb{E}_{\mathbf{I}}[\|\mathbf{f}(\mathbf{x}+\mathbf{u})-\mathbf{f}(\mathbf{x})\|]$.*

$$RINFD(\mathbf{x}) \geqslant \left( \frac{A(\mathbf{x}) - B(\mathbf{x}) - B(\mathbf{x} - \mathbf{I})}{2} \right)^2.$$

We note $A(\mathbf{x})$ can be approximated by $\text{SENS}_{\text{MAX}}(\Phi, \mathbf{f}, \mathbf{x}, r)$, which shows that $A(\mathbf{x})$ is related to the sensitivity of explanation $\Phi$ and $B(\mathbf{x})$ can be seen as the sensitivity of function $\mathbf{f}$. When the explanation is much more sensitive than the model, the robust infidelity is lower bounded by the difference between explanation sensitivity and model sensitivity, which is clearly undesirable. This suggests that we may improve the explanation sensitivity along with explanation infidelity.

# D   Some Proofs

## D.1   Proof of Proposition 2.1

*Proof.* By some simple calculation, the optimal explanation $\Phi(\mathbf{f}, \mathbf{x})^*$ can be reduced to

$$\arg \min_{\Phi(\mathbf{f}, \mathbf{x})} \mathbb{E}_{\mathbf{I}}[\|\mathbf{I}^T\Phi(\mathbf{f}, \mathbf{x}) - (\mathbf{f}(\mathbf{x}) - \mathbf{f}(\mathbf{x} - \mathbf{I}))\|^2],$$

$$= \arg \min_{\Phi(\mathbf{f}, \mathbf{x})} \int \|\mathbf{I}^T\Phi(\mathbf{f}, \mathbf{x}) - (\mathbf{f}(\mathbf{x}) - \mathbf{f}(\mathbf{x} - \mathbf{I}))\|^2 d\mu_{\mathbf{I}},$$

$$= \arg \min_{\Phi(\mathbf{f}, \mathbf{x})} \int \|\mathbf{I}^T[\Phi(\mathbf{f}, \mathbf{x}) - \int_0^1 \nabla_{\mathbf{x}}\mathbf{f}(\mathbf{x} - \mathbf{I} + t\mathbf{I})dt]\|^2 d\mu_{\mathbf{I}},$$

by setting the first order derivative to 0, and replacing $\int_0^1 \nabla_{\mathbf{x}}\mathbf{f}(\mathbf{x} - \mathbf{I} + t\mathbf{I})dt$ by $IG(\mathbf{x}, \mathbf{I})$, we obtain the first order condition for the optimal explanation $\Phi(\mathbf{f}, \mathbf{x})^*$ when $\int \mathbf{I}\mathbf{I}^T d\mu_{\mathbf{I}}$ is inversible [8],

$$\int \mathbf{I}\mathbf{I}^T(\Phi(\mathbf{f}, \mathbf{x})^* - IG(\mathbf{x}, \mathbf{I})d\mu_{\mathbf{I}} = 0,$$

$$\implies \Phi(\mathbf{f}, \mathbf{x})^* = \left( \int \mathbf{I}\mathbf{I}^T d\mu_{\mathbf{I}} \right)^{-1} \left( \int \mathbf{I}\mathbf{I}^T IG(\mathbf{x}, \mathbf{I})d\mu_{\mathbf{I}} \right). \tag{11}$$

□

### D.2 Proof of Proposition 2.3

*Proof.* When the outcome of $\mathbf{I}$ is $\epsilon \cdot e_i$ where $\epsilon \to 0$ and the infidelity is 0, we obtain $\epsilon_i \Phi_i = \mathbf{f}(\mathbf{x}) - \mathbf{f}(\mathbf{x} - \epsilon_i)$, and thus $\Phi = \nabla_{\mathbf{x}} \mathbf{f}(\mathbf{x})$. $\qquad\square$

### D.3 Proof of Propostion 2.4

*Proof.* When the outcome of $\mathbf{I}$ is $e_i \odot \mathbf{x}$ and the infidelity is 0, we obtain $\mathbf{x}_i \Phi_i = \mathbf{f}(\mathbf{x}) - \mathbf{f}(\mathbf{x}|\mathbf{x}_i = 0)$, which is occlusion-1. $\qquad\square$

### D.4 Proof of Proposition 2.5

*Proof.* We know that $\mathbf{I} = \mathbf{x} \odot \mathbf{Z}$, and $\mathbf{Z}$ is a binary vector. Let $g(\mathbf{I}) = \mathbf{f}(\mathbf{x}) - \mathbf{f}(\mathbf{x} - \mathbf{I})$, $h(\mathbf{Z}) = g(\mathbf{x} \odot \mathbf{Z})$, and $\hat{\Phi}(\mathbf{f}, \mathbf{x}) = \mathbf{x} \odot \Phi(\mathbf{f}, \mathbf{x})$, then

$$
\begin{aligned}
\text{INFD}(\Phi, \mathbf{f}, \mathbf{x}) &= \mathbb{E}_{\mathbf{I}}[\|\mathbf{I}^T \Phi(\mathbf{f}, \mathbf{x}) - [\mathbf{f}(\mathbf{x}) - \mathbf{f}(\mathbf{x} - \mathbf{I})]\|^2], \\
&= \mathbb{E}_{\mathbf{I}}[\|\mathbf{I}^T \Phi(\mathbf{f}, \mathbf{x}) - g(\mathbf{I})\|^2], \\
&= \mathbb{E}_{\mathbf{Z}}[\|\mathbf{Z}^T[\mathbf{x} \odot \Phi(\mathbf{f}, \mathbf{x})] - h(\mathbf{Z})\|^2], \\
&= \sum_{\mathbf{z}' \in (0,1)^d} (h(\mathbf{z}') - \hat{\Phi}(\mathbf{f}, \mathbf{x})^T \mathbf{z}')^2 \pi_{\mathbf{x}}(\mathbf{z}'),
\end{aligned}
\tag{12}
$$

where $\pi_{\mathbf{x}}(\mathbf{z}') \propto \frac{d-1}{\binom{d}{\|\mathbf{z}'\|_1} \|\mathbf{z}'\|_1 (d - \|\mathbf{z}'\|_1)}$. By theorem 2 in [25], the optimal $\hat{\Phi}(\mathbf{f}, \mathbf{x})$ minimizing (12) is then the Shapley value corresponding to function value h, which has the form

$$
\begin{aligned}
\hat{\Phi}(\mathbf{f}, \mathbf{x})_j &= \sum_{S \subseteq N \setminus j} \frac{(d - s_j - 1)! s_j!}{d!} [h(S \cup \{j\}) - h(S)], \\
&= \sum_{S \subseteq N \setminus j} \frac{(d - s_j - 1)! s_j!}{d!} [\mathbf{f}(h_{\mathbf{x}}(\mathbf{z}_0 - S) - \mathbf{f}(h_{\mathbf{x}}(\mathbf{z}_0 - S \cup \{j\}))], \\
&= \sum_{T \subseteq N \setminus j} \frac{(d - t_j - 1)! t_j!}{d!} [\mathbf{f}(h_{\mathbf{x}}(T \cup \{j\}) - \mathbf{f}(h_{\mathbf{x}}(T)],
\end{aligned}
$$

where $T = \mathbf{z}_0 - S \cup \{j\}$ and $s_i$ and $t_i$ are the number of non-zero elements in $S$ and $T$. This is also equal to the Shapley value corresponding the original function $\mathbf{f}(h_{\mathbf{x}}(\cdot))$. $\qquad\square$

### D.5 Proof of Theorem C.1

*Proof.* Let $\mathbf{u}_1$ be the perturbation that maximizes $A(\mathbf{x})$,

$$
\mathbf{u}_1 = \arg\max_{\|\mathbf{u}\| \leqslant r} \mathbb{E}_{\mathbf{I}}[\|\mathbf{I}^T \Phi(\mathbf{f}, \mathbf{x} + \mathbf{u}) - \mathbf{I}^T \Phi(\mathbf{f}, \mathbf{x})\|],
$$

$$
\begin{aligned}
\text{RINFD}(\Phi, \mathbf{f}, \mathbf{x}) &= \max_{\|\mathbf{u}\| \leqslant r} \mathbb{E}_{\mathbf{I}}[\|\mathbf{I}^T \Phi(\mathbf{f}, \mathbf{x} + \mathbf{u}) - (\mathbf{f}(\mathbf{x} + \mathbf{u}) - \mathbf{f}(\mathbf{x} + \mathbf{u} - \mathbf{I}))\|^2], \\
&\geqslant \mathbb{E}_{\mathbf{I}}[\|\mathbf{I}^T \Phi(\mathbf{f}, \mathbf{x} + \mathbf{u}_1), \mathbf{f}(\mathbf{x} + \mathbf{u}_1) - \mathbf{f}(\mathbf{x} + \mathbf{u}_1 - \mathbf{I})\|^2].
\end{aligned}
\tag{13}
$$

$$
\begin{aligned}
\text{RINFD}(\Phi, \mathbf{f}, \mathbf{x}) &= \max_{\|\mathbf{u}\| \leqslant r} \mathbb{E}_{\mathbf{I}}[\|\mathbf{I}^T \Phi(\mathbf{f}, \mathbf{x} + \mathbf{u}) - (\mathbf{f}(\mathbf{x} + \mathbf{u}) - \mathbf{f}(\mathbf{x} + \mathbf{u} - \mathbf{I}))\|^2], \\
&\geqslant \mathbb{E}_{\mathbf{I}}[\|\mathbf{I}^T \Phi(\mathbf{f}, \mathbf{x}), \mathbf{f}(\mathbf{x}) - \mathbf{f}(\mathbf{x} - \mathbf{I})\|^2].
\end{aligned}
\tag{14}
$$

By triangle inequality we know that

$$
\begin{aligned}
&\mathbb{E}_{\mathbf{I}}[\|\mathbf{I}^T \Phi(\mathbf{f}, \mathbf{x} + \mathbf{u}_1) - [\mathbf{f}(\mathbf{x} + \mathbf{u}_1) - \mathbf{f}(\mathbf{x} + \mathbf{u}_1 - \mathbf{I})]\|] + \mathbb{E}_{\mathbf{I}}[\|\mathbf{I}^T \Phi(\mathbf{f}, \mathbf{x}) - [\mathbf{f}(\mathbf{x}) - \mathbf{f}(\mathbf{x} - \mathbf{I})]\|] \\
&\geqslant \mathbb{E}_{\mathbf{I}}[\|\mathbf{I}^T \Phi(\mathbf{f}, \mathbf{x} + \mathbf{u}_1) - \mathbf{I}^T \Phi(\mathbf{f}, \mathbf{x})\|] - \mathbb{E}_{\mathbf{I}}[\|\mathbf{f}(\mathbf{x} + \mathbf{u}_1) - \mathbf{f}(\mathbf{x})\|] - \mathbb{E}_{\mathbf{I}}[\|\mathbf{f}(\mathbf{x} + \mathbf{u}_1 - \mathbf{I}) - \mathbf{f}(\mathbf{x} - \mathbf{I})\|], \\
&\geqslant A(\mathbf{x}) - B(\mathbf{x}) - B(\mathbf{x} - \mathbf{I}).
\end{aligned}
\tag{15}
$$

Thus, we obtain

$$\text{RINFD}(\mathbf{x}) \geqslant \frac{\mathbb{E}_\mathbf{I}[\|\mathbf{I}^T\Phi(\mathbf{f}, \mathbf{x} + \mathbf{u}_1) - [\mathbf{f}(\mathbf{x} + \mathbf{u}_1) - \mathbf{f}(\mathbf{x} + \mathbf{u}_1 - \mathbf{I})]\|^2] + \mathbb{E}_\mathbf{I}[\|\mathbf{I}^T\Phi(\mathbf{f}, \mathbf{x}) - [\mathbf{f}(\mathbf{x}) - \mathbf{f}(\mathbf{x} - \mathbf{I})]\|^2]}{2}$$

$$\geqslant \frac{\mathbb{E}_\mathbf{I}[\|\mathbf{I}^T\Phi(\mathbf{f}, \mathbf{x} + \mathbf{u}_1) - [\mathbf{f}(\mathbf{x} + \mathbf{u}_1) - \mathbf{f}(\mathbf{x} + \mathbf{u}_1 - \mathbf{I})]\|]^2 + \mathbb{E}_\mathbf{I}[\|\mathbf{I}^T\Phi(\mathbf{f}, \mathbf{x}) - [\mathbf{f}(\mathbf{x}) - \mathbf{f}(\mathbf{x} - \mathbf{I})]\|]^2}{2}$$

$$\geqslant \frac{(\mathbb{E}_\mathbf{I}[\|\mathbf{I}^T\Phi(\mathbf{f}, \mathbf{x} + \mathbf{u}_1) - [\mathbf{f}(\mathbf{x} + \mathbf{u}_1) - \mathbf{f}(\mathbf{x} + \mathbf{u}_1 - \mathbf{I})]\|] + \mathbb{E}_\mathbf{I}[\|\mathbf{I}^T\Phi(\mathbf{f}, \mathbf{x}) - [\mathbf{f}(\mathbf{x}) - \mathbf{f}(\mathbf{x} - \mathbf{I})]\|])^2}{4}$$

$$\geqslant (\frac{A(\mathbf{x}) - B(\mathbf{x}) - B(\mathbf{x} - \mathbf{I})}{2})^2.$$

The first inequality can be obtained from (13) and (14), the second inequality follows from Jensen's inequality, the third follows from AM-GM inequality, and the last can be obtained by plugging in (15). Moreover, $A(\mathbf{x})$ can be approximated as

$$A(\mathbf{x}) = \max_{\|\mathbf{u}\| \leqslant r} \mathbb{E}_\mathbf{I}[\|\mathbf{I}^T\Phi(\mathbf{f}, \mathbf{x} + \mathbf{u}) - \mathbf{I}^T\Phi(\mathbf{f}, \mathbf{x})\|]$$

$$\leqslant \max_{\|\mathbf{u}\| \leqslant r} \mathbb{E}_\mathbf{I}[\|\mathbf{I}^T\| \cdot \|\Phi(\mathbf{f}, \mathbf{x} + \mathbf{u}) - \Phi(\mathbf{f}, \mathbf{x})\|]$$

$$\leqslant \mathbb{E}_\mathbf{I}[\|\mathbf{I}\| \cdot \max_{\|\mathbf{u}\| \leqslant r} \|\Phi(\mathbf{f}, \mathbf{x} + \mathbf{u}) - \Phi(\mathbf{f}, \mathbf{x})\|]$$

$$= \mathbb{E}_\mathbf{I}[\|\mathbf{I}\|] \cdot \text{SENS}_{\text{MAX}}(\Phi, \mathbf{f}, \mathbf{x}, r).$$

$\square$

## D.6 Proof of Theorem 4.1

$$\text{SENS}_{\text{MAX}}(\Phi_k, \mathbf{f}, \mathbf{x}, r)$$

$$= \max_{\|\mathbf{u}\| \leqslant r} \|\Phi_k(\mathbf{f}, \mathbf{x} + \mathbf{u}) - \Phi_k(\mathbf{f}, \mathbf{x})\|$$

$$= \max_{\|\mathbf{u}\| \leqslant r} \|\int_\mathbf{z} [\Phi(\mathbf{f}, \mathbf{z} + \mathbf{u}) - \Phi(\mathbf{f}, \mathbf{z})] k(\mathbf{x}, \mathbf{z}) d\mathbf{z}\|$$

$$\leqslant \max_{\|\mathbf{u}\| \leqslant r} \int_\mathbf{z} \|\Phi(\mathbf{f}, \mathbf{z} + \mathbf{u}) - \Phi(\mathbf{f}, \mathbf{z})\| k(\mathbf{x}, \mathbf{z}) d\mathbf{z}$$

$$\leqslant \int_\mathbf{z} \max_{\|\mathbf{u}\| \leqslant r} [\|\Phi(\mathbf{f}, \mathbf{z} + \mathbf{u}) - \Phi(\mathbf{f}, \mathbf{z})\|] k(\mathbf{x}, \mathbf{z}) d\mathbf{z}$$

$$= \int_\mathbf{z} \text{SENS}_{\text{MAX}}(\Phi, \mathbf{f}, \mathbf{z}, r) k(\mathbf{x}, \mathbf{z}) d\mathbf{z}.$$

## D.7 Proof of Theorem 4.2

*Proof.* We first show that when $C_1 < \frac{1}{\sqrt{2}}$,

$$\max_\mathbf{u} \int_\mathbf{I} \int_\mathbf{z} \|\mathbf{I}^T\Phi(\mathbf{f}, \mathbf{z} + \mathbf{u}) - [\mathbf{f}(\mathbf{x} + \mathbf{u}) - \mathbf{f}(\mathbf{x} + \mathbf{u} - \mathbf{I})]\|^2 k(\mathbf{x}, \mathbf{z}) d\mathbf{z} d\mu_\mathbf{I}$$

$$\leqslant \frac{1}{1 - 2\sqrt{C_1}} \max_\mathbf{u} \int_\mathbf{I} \int_\mathbf{z} \|\mathbf{I}^T\Phi(\mathbf{f}, \mathbf{z} + \mathbf{u}) - [\mathbf{f}(\mathbf{z} + \mathbf{u}) - \mathbf{f}(\mathbf{z} + \mathbf{u} - \mathbf{I})]\|^2 k(\mathbf{x}, \mathbf{z}) d\mathbf{z} d\mu_\mathbf{I} \tag{16}$$

To simplify the notations, we let $\mathbf{T}_1(\mathbf{I}, \Phi, \mathbf{x}) = \mathbf{I}^T\Phi(\mathbf{f}, \mathbf{z} + \mathbf{u})$ and $\mathbf{T}_2(\mathbf{I}, \mathbf{f}, \mathbf{x}) = \mathbf{f}(\mathbf{x}) - \mathbf{f}(\mathbf{x} - \mathbf{I})$ in this proof. By a simple reorder we can get the following form.

$$\int_{\mathbf{I}}\int_{\mathbf{z}}\|\mathbf{I}^T\mathbf{T}_1(\mathbf{I},\Phi,\mathbf{z}+\mathbf{u})-\mathbf{T}_2(\mathbf{I},\mathbf{f},\mathbf{x}+\mathbf{u})\|^2 k(\mathbf{x},\mathbf{z})d\mathbf{z}d\mu_{\mathbf{I}}$$

$$\leqslant\int_{\mathbf{I}}\int_{\mathbf{z}}\|\mathbf{I}^T\mathbf{T}_1(\mathbf{I},\Phi,\mathbf{z}+\mathbf{u})-\mathbf{T}_2(\mathbf{I},\mathbf{f},\mathbf{z}+\mathbf{u})\|^2 k(\mathbf{x},\mathbf{z})d\mathbf{z}d\mu_{\mathbf{I}}$$

$$+2\int_{\mathbf{I}}\int_{\mathbf{z}}\|\mathbf{I}^T\mathbf{T}_1(\mathbf{I},\Phi,\mathbf{z}+\mathbf{u})-\mathbf{T}_2(\mathbf{I},\mathbf{f},\mathbf{x}+\mathbf{u})\|\|\mathbf{T}_2(\mathbf{I},\mathbf{f},\mathbf{z}+\mathbf{u})-\mathbf{T}_2(\mathbf{I},\mathbf{f},\mathbf{x}+\mathbf{u})\|k(\mathbf{x},\mathbf{z})d\mathbf{z}d\mu_{\mathbf{I}}.$$

$$(17)$$

By Cauchy-Schwartz inequality and plugging in (6) we then have

$$(\int_{\mathbf{I}}\int_{\mathbf{z}}\|\mathbf{I}^T\mathbf{T}_1(\mathbf{I},\Phi,\mathbf{z}+\mathbf{u})-\mathbf{T}_2(\mathbf{I},\mathbf{f},\mathbf{x}+\mathbf{u})\|\|\mathbf{T}_2(\mathbf{I},\mathbf{f},\mathbf{z}+\mathbf{u})-\mathbf{T}_2(\mathbf{I},\mathbf{f},\mathbf{x}+\mathbf{u})\|k(\mathbf{x},\mathbf{z})d\mathbf{z}d\mu_{\mathbf{I}})^2$$

$$\leqslant\int_{\mathbf{I}}\int_{\mathbf{z}}\|\mathbf{I}^T\mathbf{T}_1(\mathbf{I},\Phi,\mathbf{z}+\mathbf{u})-\mathbf{T}_2(\mathbf{I},\mathbf{f},\mathbf{x}+\mathbf{u})\|^2 k(\mathbf{x},\mathbf{z})d\mathbf{z}d\mu_{\mathbf{I}}\int_{\mathbf{I}}\int_{\mathbf{z}}\|\mathbf{I}^T\mathbf{T}_2(\mathbf{I},\mathbf{f},\mathbf{z}+\mathbf{u})-\mathbf{T}_2(\mathbf{I},\mathbf{f},\mathbf{x}+\mathbf{u})\|^2 k(\mathbf{x},\mathbf{z})d\mathbf{z}d\mu_{\mathbf{I}}$$

$$\leqslant C_1(\int_{\mathbf{I}}\int_{\mathbf{z}}\|\mathbf{I}^T\mathbf{T}_1(\mathbf{I},\Phi,\mathbf{z}+\mathbf{u})-\mathbf{T}_2(\mathbf{I},\mathbf{f},\mathbf{x}+\mathbf{u})\|^2 k(\mathbf{x},\mathbf{z})d\mathbf{z}d\mu_{\mathbf{I}})^2.$$

$$(18)$$

Therefore, by plugging (18) into (17), and when $C_1 < \frac{1}{\sqrt{2}}$ we obtain

$$\int_{\mathbf{I}}\int_{\mathbf{z}}\|\mathbf{I}^T\mathbf{T}_1(\mathbf{I},\Phi,\mathbf{z}+\mathbf{u})-\mathbf{T}_2(\mathbf{I},\mathbf{f},\mathbf{x}+\mathbf{u})\|^2 k(\mathbf{x},\mathbf{z})d\mathbf{z}d\mu_{\mathbf{I}}$$

$$\leqslant\frac{1}{(1-2\sqrt{C_1})}\int_{\mathbf{I}}\int_{\mathbf{z}}\|\mathbf{T}_1(\mathbf{I},\Phi,\mathbf{z}+\mathbf{u})-\mathbf{T}_2(\mathbf{I},\mathbf{f},\mathbf{z}+\mathbf{u})\|^2 k(\mathbf{x},\mathbf{z})d\mathbf{z}d\mu_{\mathbf{I}}.$$

$$(19)$$

We note that (19) holds for any $\mathbf{u}$, and thus (16) holds directly from (19).

We can now prove the main theorem.

$$\begin{aligned}
\text{INFD}(\Phi_k,\mathbf{f},\mathbf{x}) &= \int_{\mathbf{I}}\|\int_{\mathbf{z}}\mathbf{I}^T\Phi(\mathbf{f},\mathbf{z})k(\mathbf{x},\mathbf{z})d\mathbf{z}-[\mathbf{f}(\mathbf{x})-\mathbf{f}(\mathbf{x}-\mathbf{I})]\|^2 d\mu_{\mathbf{I}}\\
&\leqslant C_2\int_{\mathbf{I}}\int_{\mathbf{z}}\|\mathbf{I}^T\Phi(\mathbf{f},\mathbf{z})-[\mathbf{f}(\mathbf{x})-\mathbf{f}(\mathbf{x}-\mathbf{I})]\|^2 k(\mathbf{x},\mathbf{z})d\mathbf{z}d\mu_{\mathbf{I}}\\
&\leqslant\frac{C_2}{1-2\sqrt{C_1}}\int_{\mathbf{I}}\int_{\mathbf{z}}\|\mathbf{I}^T\Phi(\mathbf{f},\mathbf{z})-[\mathbf{f}(\mathbf{z})-\mathbf{f}(\mathbf{z}-\mathbf{I})]\|^2 k(\mathbf{x},\mathbf{z})d\mathbf{z}d\mu_{\mathbf{I}}\\
&=\frac{C_2}{1-2\sqrt{C_1}}\int_{\mathbf{z}}\int_{\mathbf{I}}\|\mathbf{I}^T\Phi(\mathbf{f},\mathbf{z})-[\mathbf{f}(\mathbf{z})-\mathbf{f}(\mathbf{z}-\mathbf{I})]\|^2 d\mu_{\mathbf{I}}k(\mathbf{x},\mathbf{z})d\mathbf{z}\\
&=\frac{C_2}{1-2\sqrt{C_1}}\int_{\mathbf{z}}\text{INFD}(\Phi,\mathbf{f},\mathbf{z})k(\mathbf{x},\mathbf{z})d\mathbf{z}
\end{aligned}$$
$$(20)$$

The first inequality follows from (7), and the second follow from (16). $\qquad\square$

Figure 7: More randomly chosen examples for various global saliency explanations on Imagenet, we observe that while sometimes gradient-based saliency methods do not focus on the dark objects of interest, the square removal focuses on the object more consistently, which is more faithful to the model.

Figure 8: More randomly chosen examples for various local saliency explanations on MNIST, we observe that gradient-based saliency methods are more noisy compared to NB, and the NB includes region that not only are white but also those which gives more evidence to the prediction, such as the long tails for the 3's, and the upper region of 6's.

Figure 9: More randomly chosen examples of the visualization results for various the sanity check experiment on imagenet, and we observe that Square usually focuses on random objects for the random network, while gradient-based saliency maps focus more on bright objects. Therefore, Square tends to have more diversity between explanations for original model and randomized model.

Figure 10: More randomly chosen examples presented in the human evaluation experiment. For each original example, we visualize the saliency maps obtained by four different methods with respect to two different models, namely, the model with approximated ground truth being image and that being text. We see that it is hard to correctly tell the approximated ground truth (the block where the model relies on to make its prediction) from many of the saliency maps, as they might at the same time highlight both of the blocks, or even highlight the block where the model in fact does not rely on.

## Footnotes

[8] the optimal explanation is unique when $\int \mathbf{I}\mathbf{I}^T d\mu_{\mathbf{I}}$ is invertible