[Reviews · NeurIPS 2019]

Reviewer 1



Please see above for my summary of the paper's main contributions. Originality: Incremental overall. - Infidelity: The main innovation in Definition 2.1 appears to be taking the mean square error over a distribution of perturbations. While it is nice that the optimal explanation can be found as in Proposition 2.1, it is the straightforward solution of a quadratic minimization. - Sensitivity: Definition 3.1 is closely related to local Lipschitz continuity (4) and is difficult to claim as original since it is a rather basic notion. In Section 4, the idea of smoothing explanations is not novel but the analysis leading to Theorems 4.1 and 4.2 is novel. Quality: - Below Proposition 2.1 and also in the abstract and introduction, it is claimed that the "smoothing operation [is] reminiscent of SmoothGrad". I think this claim is vague and needs justification. - Since the explanation function Phi(x) is vector-valued, \nabla_x Phi should be a matrix of partial derivatives. However before (3), the text states "norm of the sensitivity vector: || \nabla_x Phi(f(x)) ||". I think a matrix norm is required and the wording needs to change. - Section 5: It would be good for completeness to also consider the classes of perturbations in Section 2.3 to verify the claims of optimality (or at least near-optimality) in that section. - Appendix B.3: Figure 6 suggests that a good value for the SmoothGrad radius is around 1 and that the value of 0.2 mentioned in Appendix B.1 is too small. Increasing the radius may yield a fairer comparison. Significance: - I think that Section 5 presents compelling evidence, particularly of the benefit of optimizing explanations for the application, and also of smoothing explanations. In Table 1, the infidelity numbers for the optimal methods are much lower. This appears to translate into qualitatively better visualizations in Figures 1 and 2 and higher human accuracy in Table 3. Figure 4 is particularly striking. - Proposition 2.1: I think more attention should be paid to the non-invertible case to be more useful. Is it true for example that if I is deterministic, the integral of I I^T is rank-one and cannot be inverted? - The motivation for Definition 3.1 and its advantages over e.g. local Lipschitz continuity (4) are unclear. Lines 211-212 state that "the main attraction ... is that it can be robustly estimated via Monte-Carlo sampling" but I don't see why this is not true for other sensitivity measures. - While I think Theorem 4.2 is a sound result, I would have liked to see it taken further to have more impact. Can the constants C_1 and C_2 be estimated in practice? Since smoothing with a Gaussian kernel has already been proposed in the form of SmoothGrad, what is the benefit of introducing a more general kernel? For example, how does the choice of kernel affect sensitivity and fidelity? Clarity: - The writing is a little repetitive overall. As one example, I think the point about simultaneously reducing sensitivity and infidelity is repeated and emphasized in too many places. As another example, lines 65-72 in the introduction could be more focused. - In the introduction, "significant" versus "insignificant" perturbations are never really defined. I think these terms are confusing since the perturbations are conceptually different: explanations should capture changes in the prediction under perturbations of interest to the test point, while the explanations themselves should not be too sensitive to perturbations of the test point. - Line 105: Should "0" be a more general baseline value? - Line 116: A different symbol is needed for the domain of I. - Line 148: Does the circle with a dot inside mean element-wise product? - Proposition 2.3: Should all basis vectors e_i be considered to get all components of the gradient, not just individual partial derivatives? - Line 191: I do not understand the modification. - Section 3: The notation for the explanation function is not consistent. Phi(f, x), Phi(f(x)), Phi(x) are all used. - Theorem 4.1: I believe it is assumed that the kernel k(x,z) is non-negative. While this might be implied by the term "kernel", I think it's clearer to make such assumptions explicit. - Section 5: Pointers to specific sections in the appendix are needed as it is hard to navigate on its own. It would be better if some of the details in the Setup paragraph could be brought into the main text, e.g. number of images, baseline value and magnitude of noise. - Appendix B.1: Why does the optimal solution require so many (20,000) Monte Carlo samples? - Appendix B.1: Does "square size from 1x1 to 10x10", etc. mean that the perturbation square size is drawn uniformly from 1 to 10? *** Response to Author Feedback *** I thank the authors in particular for their responses regarding 1) additional "sanity check" experiments, 2) tuning the SmoothGrad radius, 3) the relationship between SmoothGrad and Proposition 2.1, and 4) the non-invertible case in Proposition 2.1. I encourage the authors to add as many of these results and clarifications to the main paper as possible. Regarding 4), my quibble is that Proposition 2.2 as written states that the completeness axiom is a necessary condition for optimal infidelity but not necessarily sufficient. It would be good to check that the two are indeed equivalent.

Reviewer 2



This paper is very clearly written, providing a thorough review of previous works and appropriately contextualizing the current work. This paper also does some important bridging work: in addition to proposes new measures of explanations, the work shows how these measures relate to previous understandings of explanations, and thus providing a thorough and more unified understanding. The empirical experiments are thorough and the human evaluations provide a good validation for the measures proposed.

Reviewer 3



I understand that definition 2.1 is new, but this is not entirely clear to me. The authors should explain what is new in definition 2.1. Has it been considered in the literature in the past? The implications of Def. 2.1 presented in sections 2.2 and 2.3 are strong and interesting. One of the key contributions of the paper addresses the question of how to modify explanations to improve sensitivity and fidelity. The authors show that they can improve both sensitivity and fidelity simultaneously. In order to judge the significance of this contribution, I would expect the authors to clearly explain the differences between fidelity and sensitivity. Fidelity is based on the effect of perturbations, whereas in lines 58-59 the authors say "It is natural to wish for our explanation to not be too sensitive, since that would entail differing explanations with minor variations in the input (or prediction values)". It is clear to me that fidelity and sensitivity are highly correlated, and I am a not sure why anyone could think that it is important to show that sensitivity and fidelity could be improved simultaneously. Sensitivity seems to be a weaker metric because it does not use information about the outputs when the inputs are perturbed, whereas infidelity uses the error. So, it is a kind of trivial that infidelity that has both perturbations and access to the outputs is consistent with sensitivity that knows about perturbations but ignores the outputs. I believe that if the authors could explain the difference between sensitivity and fidelity, and it would be possible to see importance of this contribution. It seems to me that the idea of smoothing of explanations is related to "ensembles" in machine learning. If so, the authors should mention this fact clearly and state that their explanations are in fact ensemble explanations. As long as I am familiar with explainability in machine learning, my knowledge of metrics to evaluate explanations is limited. For this reason, my questions above ask for some clarifications. The experimental results are very comprehensive, and they indicate that the methods proposed in this paper can generate explanations that are useful both with respect to a subjective evaluation by humans and quantitative metrics. I have to admit that the empirical evaluation is strong. The authors could proofread the paper to fix small errors, e.g., "to to" in line 96.

[Author Response · NeurIPS 2019]

We thank all reviewers for their constructive and helpful reviews. We will incorporate their suggestions in the final version, and in particular, expand our discussions of related work as suggested by R2. We will also add clarifications and additional experiment results discussed in this response to our final draft.

**Originality and significance of the infidelity measure (R1, R3):** We believe that the main novelty of infidelity (Definition 2.1) is the introduction of the random variable that represents the perturbation, which can be chosen by the user. Previous evaluation methods consider the perturbation to be fixed (set to some baseline value), and also consider all possible combinations of features. Definition 2.1 moreover allows us to focus on only problem specific perturbations of interest that incorporate prior knowledge into the problem, and may also be more computational feasible. Surprisingly, this new degree of freedom in setting the perturbation enables us to show that many existing explanations optimize the infidelity measure with respect to some perturbations (which also shows the importance of the additional flexibility). Additionally, we are able to introduce new explanations by simply defining a new perturbation. For instance, in the human evaluation experiment we only care about whether the model looks at the image or the caption, and we define our perturbation correspondingly, which is not possible for previous evaluation metrics for explanations.

**Comprehensive Experiments (R1):** We set up a sanity check experiment on MNIST when the perturbation follows that in SHAP (Defined in Prop.2.5), and we verify that SHAP has the lowest infidelity for this perturbation, which verifies Prop.2.5. (The infidelity for each explanation is GBP-SG: 7.0, SHAP: 2.0, Square: 3.7, Grad-SG: 6.6, GBP: 10.5, IG-SG: 5.0, IG: 6.6, Grad: 12.0). We will include a more complete version of such sanity check experiments in our final version. We also verify that the infidelity for IG and SHAP is 0 when the perturbation is a constant (Prop.2.1). The 0.2 radius for SG is the parameter that optimized infidelity score by Square, but the optimal radius for SG in MNIST for NB perturbation is around 1.0 (as in Fig.6). We redid experiments where we choose the SG radius for each setting by validation, and observe little changes in the relative results (OPT > SG > vanilla explanations with a great margin).

**Smoothing Operation and General Kernel (R1):** A smoothed explanation can be defined as $\Phi_k(\mathbf{f}, \mathbf{x}) := \int_{\mathbf{z}} \Phi(\mathbf{f}, \mathbf{z}) k(\mathbf{x}, \mathbf{z}) d\mathbf{z}$ with some probability kernel $k(\cdot, \cdot)$ (which implies $k \geq 0$ by definition), which is equivalent to Smooth-Grad when $k(\mathbf{x}, \mathbf{z})$ is a Gaussian kernel. When $k(\mathbf{x}, \mathbf{z})$ is not a probability kernel, we may write the smoothed explanation as $\Phi_k(\mathbf{f}, \mathbf{x}) := [\int_{\mathbf{z}} k(\mathbf{x}, \mathbf{z})]^{-1} \int_{\mathbf{z}} \Phi(\mathbf{f}, \mathbf{z}) k(\mathbf{x}, \mathbf{z}) d\mathbf{z}$ to make it invariant of linear scaling of the kernel. Mathematically, the optimal solution of Proposition 2.1 is a smoothed explanation where $k(\mathbf{x}, \mathbf{z})$ is replaced by $\mathbf{II}^T$, which is reminiscent of Smooth-Grad by replacing the Gaussian kernel by $\mathbf{II}^T$. In our experiments, we observe sensitivity and infidelity results are close when using Gaussian and uniform kernel, but the uniform kernel (or Truncated Normal) may be beneficial when we have a hard restriction for the input region (as Gaussian Kernel is unbounded).

**Invertible Case (R1):** When I is deterministic, the integral of $\mathbf{II}^T$ is rank-one and cannot be inverted, but being optimal with respect to the infidelity can be shown to be equivalent to satisfying the Completeness Axiom (see Proposition 2.2 where we address this case). The optimal solution then is no longer unique, since many feature attributions satisfy the Completeness Axiom. One way to achieve an unique solution is to add a non-deterministic noise to the baseline (as in noisy Baseline in l167-l172, which gives an unique explanation that satisfies a "robust Completeness" Axiom.) To enhance computational stability, we can replace inverse by pseudo-inverse, or add a small diagonal matrix to overcome the non-invertible case, which works well in experiments.

**Advantage of Definition 3.1 (R1):** While the benefit of our definition 3.1 compared to local Lipschitz continuous is not the main contribution of the paper, we point out that in certain cases, local Lipschitz continuity may be unbounded in a deep network (such as using ReLU activation function for gradient explanations, which is a common setting), but definition 3.1 is always finite given that explanation score is bounded, and thus is more robust to estimate.

**Solve challenges of Localizing (feature-based) explanations (R2):** Three major critiques of feature-based saliency maps are: (a) there is no fair way to evaluate the explanation, (b) feature-based explanations may not be faithful to the model (as shown by saliency sanity checks), and (c) explanations may not be robust to small perturbations. To address the first two challenges, our work proposes an evaluation metric (infidelity) that is more general than previous evaluation methods, allowing the user to define the perturbation of interest based on the context. In the Human Evaluation experiment, our infidelity measure is able to evaluate the quality of many explanations for this specific task and lead to an optimal explanation that improves human evaluation accuracy and passes sanity check. We also show that smoothing explanation and adversarial training (in Supplement A.3) allow us to obtain more robust explanations.

**Differences between fidelity and sensitivity (R3):** We *emphasize* that fidelity and sensitivity are **not isomorphic**. The easiest way to see this is that a constant explanation with minimum sensitivity has high infidelity. Low infidelity means that the dot product between the *fixed* explanation and the perturbation $\mathbf{I}$ is close to the function change after perturbation (so that the explanation is able to capture – with fidelity – the effect on the model function given $\mathbf{I}$), while low sensitivity means that the explanation does not change much after small perturbations to the input. Thus, while both involve perturbations, their mathematical characterizations are quite different (for fidelity perturbation is for function, and for sensitivity perturbation is for explanation). The key reason that we can reduce both sensitivity and infidelity is that sensitivity of many explanations (with dot product with $\mathbf{I}$) is higher than prediction sensitivity (which is observed [11]), which makes C1 (in l249) small and leads to a lower upperbound in Theorem 4.2. One relationship between sensitivity and infidelity is shown in Appendix C, where we show that "a robust version infidelity will be large if sensitivity of explanation is much larger than prediction sensitivity". Again, this relationship only holds when sensitivity of explanation is (much) larger than prediction sensitivity. Therefore, we emphasize that while we can improve infidelity and sensitivity, this is not an universal statement and the relationship between sensitivity and fidelity is non-trivial.

[Meta-Review · NeurIPS 2019]

Reviewers agree that this paper contains valuable ideas and evaluations. Authors response on adding sanity check and others (R1's comments) has increased our confidence. Please carefully review remaining comments (e.g., R2's comment on benefits of non-feature based interpretability methods) for your final version.